

# A higher-order topological twist on cold-atom SO(5) Dirac fields

Alejandro Bermudez[1], Daniel González-Cuadra[2,3] and Simon Hands[4]

**1** Instituto de Física Teórica, UAM-CSIC, Universidad Autónoma de Madrid,
Cantoblanco, 28049 Madrid, Spain
**2** Institute for Theoretical Physics, University of Innsbruck, 6020 Innsbruck, Austria
**3** Institute for Quantum Optics and Quantum Information
of the Austrian Academy of Sciences, 6020 Innsbruck, Austria
**4** Department of Mathematical Sciences, University of Liverpool,
Liverpool L69 3BX, United Kingdom

## Abstract

Ultracold Fermi gases of spin-3/2 atoms provide a clean platform to realise SO(5) models of 4-Fermi interactions in the laboratory. By confining the atoms in a two-dimensional Raman lattice, we show how this system can be used as a flexible quantum simulator of Dirac quantum field theories (QFTs) that combine Gross-Neveu and Thirring interactions with a higher-order topological twist. We show that the lattice model corresponds to a regularization of this QFT with an anisotropic twisted Wilson mass. This allows us to access higher-order topological states protected by a discrete SO(5) group, a remnant of the continuous rotational symmetry of the 4-Fermi interactions that is not explicitly broken by the lattice discretization. Using large-$N$ methods, we show that the 4-Fermi interactions lead to a rich phase diagram with various competing fermion condensates. Our work opens a route for the implementation of correlated higher-order topological states with tunable interactions that has interesting connections to non-trivial relativistic QFTs of Dirac fermions in $D = 2 + 1$ dimensions.

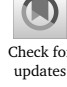

# 1   Introduction

Symmetry plays a primary role in our most fundamental theories of Nature. So far, all forms of matter observed in the laboratory can be ultimately described by the standard model [1], a relativistic quantum field theory (QFT) that contains Dirac fermions locally coupled to both scalar and gauge bosons, and is invariant under Lorentz transformations. The role of symmetry goes beyond this relativistic invariance, as the specific form of the local fermion-boson couplings is dictated by the invariance of the QFT under various groups of local, so-called gauge, symmetries [2]. Additionally, there are also global symmetries that leave the standard model invariant. These symmetries can be spontaneously broken at a certain energy scale, such that the vacuum of this QFT displays a certain order parameter that is no longer invariant under the action of the symmetry. An example of great relevance in high-energy physics is that of chiral symmetry breaking [3], which leads to a so-called chiral condensate, and accounts for most of the mass of the matter in our universe. In the early days of quantum chromodynamics (QCD) [1], various effective QFTs leading to mass generation by chiral symmetry breaking were explored, including 4-Fermi QFTs such as the Nambu-Jona-Lasinio [3–6] and Gross-Neveu [7–11] models. These QFTs describe self-interacting Dirac fermions with different quartic interactions and, moreover, can be defined in various spacetime dimensions, while showing analogies with their higher-dimensional non-Abelian counterparts [12, 13].

   Besides serving as effective models that can capture some of the phenomenology observed at particle colliders qualitatively, Dirac QFTs also appear in condensed matter and in atomic molecular and optical (AMO) physics, where one can indeed test their predictions quantitatively. Systems such as graphene [14, 15], topological insulators [16, 17], and ultracold atoms in optical lattices [18, 19], are clear examples that provide a neat playground for low-dimensional QFTs of fermionic Dirac matter [20, 21]. In addition, Weyl semi-metals provide instances where the low-energy physics is captured by Dirac-type QFTs in $D = 3 + 1$ dimensions [22], and can also be implemented with ultracold atoms [23]. We note that, away from $D = 1 + 1$ dimensions, it is not straightforward to find experimental setups which, in spite of being highly non-relativistic, can be accurately described by Lorentz-invariant effective QFTs. Moreover, low-dimensional Dirac QFTs can present a number of non-trivial properties for $D < 4$ [24–26], and are still the subject of active research on their own. A particular example is that of Dirac QFTs in $D = 2 + 1$ dimensions which, in the presence of self-interactions

such as the above Gross-Neveu 4-Fermi term [27,28] or variants thereof [29], can give rise to strong correlations and novel critical phenomena. This type of 4-Fermi model leads to different phase transitions that are no longer characterised by the above chiral condensate but, instead, require finding other symmetry-breaking processes with their associated order parameters. These generalised 4-Fermi models have been the subject of renewed interest in recent years (see the reviews [30,31] and references therein).

In this work, we are interested in a novel type of models of synthetic Dirac QFTs that can be implemented with ultracold atoms [32]. A rather unique possibility of these systems is that, in addition to the emerging Lorentz invariance, one can actually design other symmetries experimentally, both local and global ones. These platforms can thus be used as analogue quantum simulators [33–37] for a specific QFT of interest, allowing us to test theoretical predictions in a controllable experimental environment, entering regimes that cannot be simulated using current analytical or numerical methods, such as real-time dynamics or finite fermion densities [38,39]. In particular, we focus on spin-3/2 neutral atoms at ultra-cold temperatures, as their $s$-wave scattering leads to 4-Fermi terms that naturally yield a large symmetry group of SO(5) transformations [40,41]. When these cold atoms are loaded on standard optical lattices, one obtains a non-relativistic SO(5) Hubbard model [42] that is interesting in its own right [40,43,44], and leads to various interesting phases even for $D = 1 + 1$ dimensions [45–48]. In fact, the role of SO(5) symmetry in condensed-matter systems is broader, as it also plays a role in theories of competing magnetism and superconductivity [49,50].

We show in our work that, by including additional Raman beams that interfere with the standing wave underlying the above optical lattice, we can enter a new regime in which the SO(5) 4-Fermi model corresponds to a specific discretization of a relativistic QFT of self-interacting Dirac fermions. More specifically, we demonstrate how this lattice regularization, which in general breaks explicitly the continuous rotational symmetry, can actually preserve a discrete SO(5) rotation, and provide a neat route to explore correlation effects in high-order topological insulators (HOTIs) [51–53]. In particular, this lattice discretization corresponds to an anisotropic version of twisted-mass Wilson fermions which, as we show, leads to flat bands and strictly-localised zero-energy corner modes protected by the action of a subgroup SO(5), which we henceforth refer to as a 'discrete SO(5) rotation'. These anomalous corner modes are a boundary manifestation of a non-trivial topological invariant in the bulk [54], which connects to the phenomenology explored for other lattice models of higher-order topological states. In these models, studying the effects brought up by fermion-fermion interactions has seen an increase of activity recently [55–72], but still remains largely unexplored in comparison to their first-order counterparts (see e.g. [73–76]).

We contribute to this line of research by exploring the effect that the SO(5) 4-Fermi interactions have on the aforementioned Wilson-fermion HOTI. In connection to the fermion condensate associated to chiral symmetry breaking in high-energy physics, we show that our model accounts for a competition between various possible condensation channels and that, at sufficiently strong interactions, the HOTI phase gives way to a pseudo-scalar fermion condensate where the discrete SO(5) rotational symmetry that protects the HOTI gets spontaneously broken. We present a non-perturbative account of this phenomenon based on the large-$N$ limit of this 4-Fermi QFT, where one considers $N$ flavours of the Dirac fermions coupled by the quartic SO(5) interactions. By resuming the leading-order Feynman diagrams for $N \to \infty$, we calculate the effective potential, and perform a minimization that allows us to infer the values of various condensates. Moreover, this large-$N$ techniques can be readily used to obtain an estimate for the many-body topological invariant, allowing us to chart the entire phase diagram of the model. We show that, in addition to the aforementioned condensates, correlated HOTIs and trivial band insulators appear, which can be connected by topological and more standard second-order phase transitions. Since the model studied can be realised in with spin-3/2 neu-

tral atoms in Raman optical lattices, possible future experiments could test these predictions and their connection to non-trivial strongly-coupled QFTs.

# 2 Euclidean 4-Fermi field theories with a twist

In this section, we introduce our model of interacting Dirac fields in $D = 2 + 1$ dimensions, which is motivated by a specific Kaluza-Klein-like dimensional reduction. We also discuss a non-standard lattice regularization that will allow us to study the non-perturbative phenomena induced by fermion-fermion interactions on higher-order topological groundstates.

## 2.1 Dimensional reduction and SO(5) ↦ SO(3)

Our model of self-interacting Dirac matter is built from a relativistic QFT of fermions with rotationally-invariant 4-Fermi interactions. As discussed in Appendix A, the partition function of this QFT can be written as a path integral [77] over two independent Grassmann spinors $\psi(x), \overline{\psi}(x)$ [78], which represent the Dirac fermions in a 3-dimensional Euclidean spacetime with imaginary time $x = (\tau, \boldsymbol{x})$, where $\boldsymbol{x} = (x_1, x_2)$. The path integral is expressed in terms of an Euclidean action that contains two terms $S = S_0 + S_{\text{int}}$. The first one describes free Dirac fermions with two possible mass terms

$$S_0 = \int \mathrm{d}^3 x \, \overline{\psi}(x) \Big( \gamma^\mu \partial_\mu + \mathrm{i} m_1 \gamma^3 + \mathrm{i} m_2 \gamma^5 \Big) \psi(x), \tag{1}$$

where $\mu \in \{0, 1, 2\}$ labels the spacetime coordinates, $\partial_\mu = \partial/\partial x^\mu$. Here, $m_1, m_2$ are the corresponding bare masses, which will be latter connected to a mass twisting, and we note that Einstein's convention of repeated index summation and natural units $\hbar = c = 1$ are used in the following. The set of gamma matrices $\{\gamma^0, \gamma^1, \gamma^2, \gamma^3, \gamma^5\}$ fulfill $\{\gamma^a, \gamma^b\} = \gamma^a \gamma^b + \gamma^b \gamma^a = 2\delta^{a,b} \mathbb{1}_{d_s}$, which can only be satisfied by considering Grassmann fields with $d_s = 4$ spinor components. We recall that, in an Euclidean metric, these gamma matrices are all Hermitian, and can be defined via tensor products of operators within the Pauli basis $\{\mathbb{1}, \sigma^x, \sigma^y, \sigma^z\}$ [79, 80]. Although the specific choice of gamma matrices is arbitrary at the level of the QFT (1), the implementation based on spin-3/2 cold-atom gases discussed in Sec. 3.2 fixes

$$\gamma^0 = \sigma^y \otimes \mathbb{1}_2, \qquad \gamma^1 = \sigma^x \otimes \sigma^x, \qquad \gamma^2 = \sigma^x \otimes \sigma^y. \tag{2}$$

In addition, the remaining gamma matrices are also fixed as

$$\gamma^3 = \sigma^x \otimes \sigma^z, \qquad \gamma^5 = -\sigma^z \otimes \mathbb{1}_2. \tag{3}$$

As discussed in Appendix A, this set of matrices actually forms a reducible representation of the Clifford algebra for the underlying 3-dimensional spacetime. These gamma matrices can be used to define the generators of Lorentz transformations, which correspond to a spinor representation of the SO(3) rotations in the Euclidean metric. We remark that such Lorentz transformations could also be generated using an irreducible representation of the Clifford algebra, which would only require using two-component spinors [79,80]. However, this choice would not permit introducing the two independent mass terms in Eq. (1), as the spacetime gamma matrices would already exhaust all the possible mutually anti-commuting Hermitian matrices, e.g. $\gamma^0 = \sigma^z, \gamma^1 = \sigma^x, \gamma^2 = \sigma^y$. In this case, there can only be a single mass term $m_0 \overline{\psi} \psi$ that breaks parity [28]. As we will see, having two anti-commuting masses (1) plays a key role in our work.

As discussed in detail in Appendix B, a different perspective motivating the choice of the action (1) is that the reducible representation (2)-(3) is the result of a Kaluza-Klein-type compactification. In the original Kaluza-Klein context [81], gravity and electrodynamics in $D = 3 + 1$

dimensions were shown to result from the compactification of an extra dimension in a 5-dimensional theory of pure gravity. In the current context (1), the situation is much simpler, as one only needs to consider a QFT of Dirac fermions in a higher 5-dimensional spacetime coupled to background fields [80,82], which will be responsible for the two mass terms. In the larger spacetime, SO(5) symmetry is manifest in the action, and can also be used to understand the structure of the Lorentz-invariant 4-Fermi self-interactions, as we now discuss. Considering the reducible representation of the Clifford algebra, we can define SO(5)-invariant 4-Fermi interactions as follows

$$S_{\text{int}} = \int \mathrm{d}^3 x \, \frac{g^2}{2} \Big( -J_\mu J^\mu + (\overline{\psi}\gamma^3\psi)^2 + (\overline{\psi}\gamma^5\psi)^2 - (\overline{\psi}\psi)^2 \Big), \tag{4}$$

where we have introduced the Euclidean fermion current as $J^\mu = \mathrm{i}\overline{\psi}\gamma^\mu\psi$. This action can again be interpreted from the perspective of dimensional reduction, where the first five terms can be written as the squared norm of a higher-dimensional vector whose components are fermion bilinears. Therefore, the norm of the vector is conserved under the SO(5) Lorentz transformations (see Appendix B). In addition, the last term in Eq. (4) is a scalar under these SO(5) spatial rotations, and thus also remains invariant. From the perspective of the higher-dimensional parent theory, these quartic terms correspond to a linear combination of the standard Gross-Neveu and Thirring [83,84] interactions. This is very different from the interactions allowed by an irreducible representation, which can all be reduced to a single Gross-Neveu quartic term. On the other hand, for a reducible representation, the physics of Gross-Neveu and Thirring QFTs in $D = 2+1$ turns out to be very different [85]. Therefore, working with reducible gamma matrices leads to a richer dimensionally-reduced QFT with more interaction channels than those allowed by an irreducible representation. The structure of Eq. (4) will yield a competition of various fermion condensates with the aforementioned topological groundstates.

## 2.2 Twisted Wilson fermions and SO(3) breakdown

After setting up the continuum QFT in Eqs. (1)-(4), we can now discuss how higher-order topological [51–53] and trivial groundstates may arise as the result of a non-standard lattice regularization. We introduce this regularization by starting from the connection of Wilson fermions [86] with first-order topological insulators [87,88], and then highlight the "twist" that is required to move to higher-order topology.

Let us start by noting that all of these regularizations are related to the problem of fermion doubling [89,90] in lattice field theories [91]. In particular, we shall consider a non-zero lattice spacing $a$ along the spatial directions, such that $\boldsymbol{x} = \sum_j a(n_j - N_j/2)\mathbf{e}_j = a(\boldsymbol{n} - \boldsymbol{N}/2)$ with $\boldsymbol{n} \in \Lambda_s = \mathbb{Z}_{N_1} \times \mathbb{Z}_{N_2}$ and $\boldsymbol{N} = (N_1, N_2)$ contains an even number of lattice sites per axis, leading to $N_s = N_1 N_2$ as the total number of lattice sites (see Fig. 1(a)). The spatial derivatives in Eq. (1) are substituted by finite differences, while the Euclidean time remains continuous $\tau \in \mathbb{R}$. This asymmetric treatment of the spacetime will allow us to make a direct connection with the Hamiltonian approach to field theories in section 3. The fermion doubling can be understood by writing the free action (1) after this regularization which, in momentum space, reads

$$S_0 = \int_k \overline{\psi}(k) \big( \gamma^\mu \hat{k}_\mu + \mathrm{i}m_1\gamma^3 + \mathrm{i}m_2\gamma^5 \big) \psi(k), \tag{5}$$

where $k = (k^0, \boldsymbol{k})$ is the three-momentum, an we have introduced the short-hand notation $\int_k := \frac{1}{(N_s a)^2}\sum_{\boldsymbol{k}\in\text{BZ}} \int \frac{\mathrm{d}k^0}{2\pi}$. In the expression above, we have defined $\hat{k}_0 = k_0 \in \mathbb{R}$ using the zero-temperature limit of the Matsubara frequencies [77]. Additionally, the spatial components of the momentum $\hat{\boldsymbol{k}}$ are related to the corresponding crystal momenta $\boldsymbol{k}$ via

$$\hat{k}_1 = \frac{1}{a}\sin(k_1 a), \qquad \hat{k}_2 = \frac{1}{a}\sin(k_2 a). \tag{6}$$

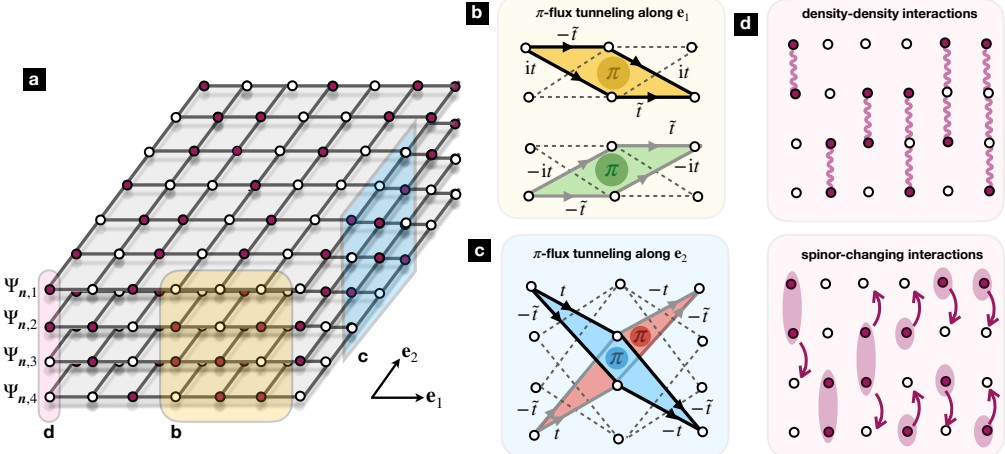

Figure 1: **Multi-layer scheme for the lattice regularization of the SO(5) 4-Fermi QFT: (a)** In the Hamiltonian formulation, the spatial directions are discretized on a square lattice, and the spinor components can be represented as four-layer model. The yellow region in the inset **(b)** describes the discretization of the kinetic and twisted Wilson mass along the $x$ axis. The layers only get coupled in pairs, and can be interpreted as a distribution of rhomboid plaquettes pierced by $\pi$-fluxes corresponding to a pair of decoupled Creutz ladders. The blue region in the inset **(b)** describes the discretization of the kinetic and twisted Wilson mass along the $y$ axis. This tunnelling couples all four layers, and also has underlying $\pi$-fluxes which, together with those of bf (b) lead to flat bands. The purple region in the inset **(d)** represents the SO(5) 4-Fermi interactions, which include Hubbard density-density interactions among all possible inter-layer pairings (upper panel), and also spin-changing collisions that can be interpreted as inter-layer pair tunnelling processes.

We recall that, as a consequence of the lattice regularization, the crystal momenta are quantised within the first Brillouin zone $\boldsymbol{k} \in \mathrm{BZ}$, namely $k_j = -\pi/a + 2\pi n_j/aN_j$ with $n_j \in \mathbb{Z}_{N_j}$. By a Taylor expansion $\boldsymbol{k} \mapsto \boldsymbol{k}_{\mathrm{D},\ell} + \boldsymbol{k}$ for $|\boldsymbol{k}| \ll \Lambda_{\mathrm{c}} = \pi/a$, around any of the four Dirac points

$$\boldsymbol{k}_{\mathrm{D},\ell} = \frac{\pi}{a}(\ell_1, \ell_2), \qquad \boldsymbol{\ell} \in \mathbb{Z}_2 \times \mathbb{Z}_2, \tag{7}$$

one finds that the long-wavelength action stemming from Eq. (5) coincides with that of a massless Dirac fermion (1). Altogether, the long-wavelength infra-red (IR) features are governed by $N_{\mathrm{D}} = 4$ Dirac fermions instead of one. Typically, one refers to $\boldsymbol{\ell} \in \{(0,1),(1,0),(1,1)\}$ as the spurious fermion doublers, each of which has a different emergent chirality $\gamma_{\ell}^5 = (-1)^{\ell_1+\ell_2}\gamma^5$. The presence of these spurious doublers can change the physics considerably, specially when adding further interactions such as those in Eq. (4).

The idea of K. Wilson to cope with fermion doubling [86] was to add a momentum-dependent *Wilson mass*, which sends these doublers to the ultra-violet (UV) cutoff scale of the QFT. Hence, they become very massive and are not expected to interfere with the low-energy physics of the remaining Dirac fermion at $\boldsymbol{\ell} = (0,0)$. In Appendix C, we show that the standard Wilson regularization amounts to setting $m_2 \mapsto \overline{m}_2 = 0$ and $m_1 \mapsto \overline{m}_1(\boldsymbol{k})$ in Eq. (C.1), which leads to two copies of a Chern insulator [92–96]. In this sense, the use of a reducible representation of the gamma matrices is rather trivial, and we could have obtained the same topological features using an irreducible representation and two-component spinors.

Note, however, that this regularization only exploited one of the mass terms in Eq. (5), and followed the Wilson prescription verbatim. In this work, we explore a different non-standard

regularization which, although having a similar effect on the doublers, leads to very different manifestations of topology. We consider an *anisotropic twisted Wilson mass* regularization $S = S_0^{\text{TM}} + S_{\text{int}}$, in which the free part is expressed in momentum space as

$$S_0^{\text{TM}} = \int_k \overline{\psi}(k)\left(\gamma^\mu \hat{k}_\mu + im_1(\boldsymbol{k})\gamma^3 + im_2(\boldsymbol{k})\gamma^5\right)\psi(k). \tag{8}$$

This contains the anisotropic twisted Wilson mass

$$
\begin{aligned}
m_1(\boldsymbol{k}) &= m_1 + \frac{r}{a}\left(1 - \cos(k_1 a)\right), \\
m_2(\boldsymbol{k}) &= m_2 + \frac{r}{a}\left(1 - \cos(k_2 a)\right),
\end{aligned} \tag{9}
$$

where $r \in [0,1]$ is the analogue of the Wilson parameter in the standard regularization of App. C. The interacting part of the action $S_{\text{int}}$ is defined by considering the 4-Fermi terms in Eq. (4), but discretizing the spatial coordinates such that the integral becomes a shorthand for $\int \mathrm{d}^3 x := a^2 \sum_{\boldsymbol{n}\in\Lambda_s} \int \mathrm{d}\tau$.

Twisted-mass Wilson fermions have been previously considered in lattice gauge theories [97–105] (see [106] for a detailed account), although the twisting procedure is very different from the one considered in our work. In order to understand the differences, let us describe mass twisting in a broader context. We start by focusing on the standard situation, which involves even-dimensional spacetimes such as $D = 3 + 1$. As discussed in Appendix A, the usual mass twisting follows from an axial rotation of angle $\theta$ on the standard Dirac mass term $m\overline{\psi}\psi$, leading to a pair of anti-commuting mass matrices proportional to $m_1 = m\cos\theta, m_2 = m\sin\theta$ (A.7). The new mass term for $\theta \neq 0$ breaks parity (A.9) explicitly, unless the rotation/twisting angle is promoted to a pseudo-scalar axion field $\theta \mapsto \theta(\tau, \boldsymbol{x})$ [107–109]. Let us now consider this mass twisting for a Wilson-fermion lattice regularization, which upgrades the parity-invariant mass to a momentum-dependent one $m_1 \mapsto \overline{m}_1(\boldsymbol{k})$ similar to Eq. (C.1), but accounting for the larger spatial dimensions. This Wilson term yields again different masses for each of the Dirac points, and can lead to a non-zero topological invariant. From the perspective of first-order topological insulators [94], a non-zero twisted mass $m_2 > 0$ breaks both parity and time-reversal, making the topological invariant no longer quantised [110] which to connects to axion electrodynamics [111].

All of these effects require having an odd number of Dirac points with negative Wilson masses which, in the context of gauge theories such as QCD, would shift the value of the vacuum theta angle. When the goal is to recover the continuum limit of QCD, it is sensible to avoid this possibility unless one is using a discretization based on domain-wall fermions with an extra spatial dimension [112–114]. Accordingly, the works on lattice QCD with Wilson fermions typically work in regions of parameter space that are close to a critical line in order to recover a continuum limit, but making sure that these topological contributions to the theta angle are zero. The motivation to include a mass twisting is then completely different from the above discussion. When trying to improve the lattice regularization to achieve a faster convergence to the continuum [115–117], the approaches based on Wilson fermions [118, 119] can take advantage of the mass twisting [98, 99]. In fact, as first shown in [100], by working with a maximal twist angle $\theta = \pi/2$, one can find an automatic $\mathcal{O}(a)$ improvement that only requires controlling a single parameter of the model.

If we now move back to our $D = 2 + 1$ dimensions, as noted above, an irreducible representation of the gamma matrices would forbid a mass twisting. On the other hand, our reducible representation (2)-(3) permits additional mass terms (1), yielding a a 3-dimensional version of the mass twisting. Still, we remark that this twisted Wilson mass would still differ from our anisotropic mass twisting in Eq. (9). As discussed in [106], the usual mass twisting for

Wilson fermions can be transformed to a physical basis where, rather than the bare masses, one rotates the Wilson term responsible for the momentum dependence of the mass. Exploring angles different from the full twist, e.g. $\theta = \pi/4$, would bring us closer to the type of Wilson mass twisting of Eq. (9). Yet, there is a fundamental difference, the Wilson mass twisting considered in lattice gauge theories is always isotropic. We are instead considering that the dependence on momentum of the twisted Wilson masses (9) is highly anisotropic: $m_1(\boldsymbol{k})$ only depends on $k_1$, and $m_2(\boldsymbol{k})$ on $k_2$. In turn, as will become clear below, this means that the real-space Wilson term is anisotropic; it is different when the fermions tunnel to neighbouring sites along each of the two spatial directions. The other important difference is that, instead of considering a mass twisting that combines $\overline{\psi}\psi$ and $\overline{\psi}i\gamma^5\psi$, we are admixing $\overline{\psi}i\gamma^3\psi$ and $\overline{\psi}i\gamma^5\psi$ terms, both of which are parity invariant in $D = 2+1$ dimensions. This will be very important to get a model invariant under a discrete SO(5) rotation, which will protect the higher-order topological state.

In the following section, we will show in detail how the groundstate of the free lattice action can be characterised by a topological invariant which is, however, distinct from that (C.6) of the Chern insulators discussed above. Indeed, as shown in Sec. 3, the groundstate in this case corresponds to a higher-order topological insulator (HOTI). The bulk-boundary correspondence [120] leads to a boundary manifestation that differs from the edge states of Chern insulator, as we find zero-energy states that are only localised in the corners of the spatial lattice. To make this connection clearer, we start by introducing a Hamiltonian version of this QFT.

## 3   Cold-atom Hamiltonian field theory

In this section, we present the Hamiltonian of the above Euclidean field theory (8), which will be useful when discussing the HOTI, and a possible cold-atom implementation.

### 3.1   The Creutz-Hubbard multi-layer

Since our discretization keeps the imaginary time continuous, one can also describe the system through a Hamiltonian lattice field theory by rotating back to real time $\tau \mapsto -ix^0$. In the Hamiltonian formulation [1, 121], one works with field operators instead of Grassmann variables. We thus define $\Psi_{\boldsymbol{n}} = (\Psi_{\boldsymbol{n},0}, \Psi_{\boldsymbol{n},1}, \Psi_{\boldsymbol{n},2}, \Psi_{\boldsymbol{n},3})^{\mathrm{t}}$ and $\Psi_{\boldsymbol{n}}^\dagger = (\Psi_{\boldsymbol{n},0}^\dagger, \Psi_{\boldsymbol{n},1}^\dagger, \Psi_{\boldsymbol{n},2}^\dagger, \Psi_{\boldsymbol{n},3}^\dagger)$ in terms of fermionic creation-annihilation operators defined on the lattice sites (see Fig. 1 **(a)**), which are supplemented with the following equal-time anti-commutation relations

$$\left\{ \Psi_{\boldsymbol{n}_1,\sigma_1}, \Psi_{\boldsymbol{n}_2,\sigma_2}^\dagger \right\} = \frac{1}{a^2}\, \delta_{\boldsymbol{n}_1,\boldsymbol{n}_2} \delta_{\sigma_1,\sigma_2}. \tag{10}$$

In a Minkowski spacetime, the adjoint is no longer independent but, instead, related to the creation operators $\overline{\Psi}_{\boldsymbol{n}} = \Psi_{\boldsymbol{n}}^\dagger \gamma^0$. The Hamiltonian operator $H$ governing the dynamics of these fields can be found from the partition function discussed in Appendix A, recalling that the basis of fermionic coherent states $\Psi_{\boldsymbol{n}} |\psi_{\boldsymbol{n}}\rangle = \psi_{\boldsymbol{n}} |\psi_{\boldsymbol{n}}\rangle$, $\langle\psi_{\boldsymbol{n}}| \overline{\Psi}_{\boldsymbol{n}} = \langle\psi_{\boldsymbol{n}}| \overline{\psi}_{\boldsymbol{n}}$ is used to write the partition function $Z = \mathrm{Tr}\{e^{-\beta H}\}$ as a path integral over the Grassmann fields $\overline{\psi}_{\boldsymbol{n}}, \psi_{\boldsymbol{n}}$ [77]. The identified operator can be written as the sum of two terms $H = H_0^{\mathrm{TM}} + H_{\mathrm{int}}$. The free term $H_0^{\mathrm{TM}}$ is obtained by the discretization of the spatial derivatives of Eq. (1) in terms of finite differences, which leads to tunnelling terms between nearest neighbours. Additionally, the anisotropic twisted Wilson mass is also realized by including tunnelling terms that give momentum-dependence to the local masses in (1) according to Eq. (9). Altogether, this leads to the quadratic lattice Hamiltonian

$$H_0^{\mathrm{TM}} = a^2 \sum_{\boldsymbol{n}\in\Lambda_s} \sum_{j=1,2} \left( \left( \Psi_{\boldsymbol{n}}^\dagger \mathbb{T}_j \Psi_{\boldsymbol{n}+\mathbf{e}_j} + \mathrm{H.c.} \right) + \Psi_{\boldsymbol{n}}^\dagger \mathbb{M}_j \Psi_{\boldsymbol{n}} \right), \tag{11}$$

where we have introduced the tunnelling matrices

$$\mathbb{T}_1 = -\mathrm{i}t\alpha^1 - \tilde{t}\alpha^3, \qquad \mathbb{T}_2 = -\mathrm{i}t\alpha^2 - \tilde{t}\alpha^5. \tag{12}$$

These depend on the following tunnelling strengths

$$t = \frac{1}{2a}, \qquad \tilde{t} = \frac{r}{2a}, \tag{13}$$

which should not be confused with real time. In the following, we shall work with Euclidean time $\tau$, such that there is no potential confusion. We also use the Dirac $\alpha$-matrices

$$\begin{aligned}
\alpha^1 &= \mathrm{i}\gamma^0\gamma^1 = \sigma^z \otimes \sigma^x, & \alpha^2 &= \mathrm{i}\gamma^0\gamma^2 = \sigma^z \otimes \sigma^y, \\
\alpha^3 &= \mathrm{i}\gamma^0\gamma^3 = \sigma^z \otimes \sigma^z, & \alpha^5 &= \mathrm{i}\gamma^0\gamma^5 = \sigma^x \otimes \mathbb{1}_2.
\end{aligned} \tag{14}$$

Note that the above matrices are still expressed in terms of products of the Euclidean gamma matrices in Eqs. (2)-(3). In Minkowski spacetime, it is customary to work with $\{\hat{\gamma}^a, \hat{\gamma}^b\} = 2\eta^{a,b}\mathbb{1}_{d_s}$, where $\eta = \mathrm{diag}\{1, -1, \cdots, -1\}$ is the metric. Using the prescription $\hat{\gamma}^0 = \gamma^0$, $\hat{\gamma}^1 = \mathrm{i}\gamma^1$, $\hat{\gamma}^2 = \mathrm{i}\gamma^2$, $\hat{\gamma}^3 = \mathrm{i}\gamma^3$, $\hat{\gamma}^5 = \gamma^5$, one recovers the standard conventions for the Hamiltonian lattice field theory of Dirac fermions [1]. Using the standard definitions of the Dirac $\alpha$ and $\beta$ matrices, we also find

$$\beta = \gamma^0 = \sigma^y \otimes \mathbb{1}_2, \qquad \alpha^j = \beta\hat{\gamma}^j, \tag{15}$$

where the later coincide with the expressions in Eq. (14).

In the above lattice Hamiltonian (11), we have also introduced the mass matrices

$$\mathbb{M}_1 = \tilde{m}_1\alpha^3, \qquad \mathbb{M}_2 = \tilde{m}_2\alpha^5, \tag{16}$$

which are expressed in terms of the following parameters

$$\tilde{m}_1 = m_1 + 2\tilde{t}, \qquad \tilde{m}_2 = m_2 + 2\tilde{t}. \tag{17}$$

In addition to this quadratic Hamiltonian (11), the 4-Fermi terms in Eq. (4) lead directly to the quartic interactions

$$H_{\mathrm{int}} = a^2 \sum_{\boldsymbol{n} \in \Lambda_s} \frac{g^2}{2} \left( (\Psi_{\boldsymbol{n}}^\dagger \Psi_{\boldsymbol{n}})^2 - \left( (\Psi_{\boldsymbol{n}}^\dagger \boldsymbol{\alpha} \Psi_{\boldsymbol{n}})^2 - (\Psi_{\boldsymbol{n}}^\dagger \beta \Psi_{\boldsymbol{n}})^2 \right) \right). \tag{18}$$

In the next section, we will see how this specific interaction emerges naturally when considering spin-3/2 Fermi gases tightly-confined by optical potentials [40, 41]. We will argue that this connection fixes the choice of the $\alpha$ and $\beta$ matrices to those in Eqs. (14)- (15), and thus forces the choice of gamma matrices (2)-(3) in our 4-Fermi QFT in Eqs. (1) and (4).

In the Hamiltonian formulation, the discussion of the SO(5) invariance must be revisited in light of the definition of the adjoint operator below Eq. (10). In fact, the Euclidean SO(5) Lorentz invariance must now be described in terms of SO(1,4) Lorentz transformations, where the boosts do not admit a unitary spinor representation [1]. One could define a completely analogous Kaluza-Klein compactification, where the above Hamiltonians arise from a 5-dimensional parent model regularised on a lattice. In analogy to the Euclidean action, the continuum limit of the dimensionally-reduced model is expected to recover the lower-dimensional SO(1,2) invariance. On the other hand, we are not only interested in the continuum limit, but also in the HOTI phases of the theory where one can go beyond this continuum emergent symmetry. From this perspective, we should look for other transformations, including discrete spatial transformations, which correspond to exact symmetries of the full lattice model in (11) and (18).

As depicted in Fig. 1 **(a)**, the non-interacting Hamiltonian (11) can be interpreted as a multi-layer fermionic model with both intra- and inter-layer tunnellings. An aspect that will be important in our analysis below is that there are certain background $\pi$-fluxes that dress the tunnelling along certain plaquettes involving the inter-layer synthetic dimensions (see Fig. 1 **(b)-(c)**). These fluxes lead to flat-band regimes, and a generalization of the so-called Aharonov-Bohm cages [122], which become very useful to understand the bulk-boundary correspondence. In this regard, our model can be considered as a higher-dimensional multi-layer version of the Creutz ladder [123–129]. Moreover, the quartic interactions (18) are purely local, and can thus be interpreted as a Hubbard-like interaction. For $D = 1 + 1$ dimensions, the Hubbard interaction maps exactly onto a Gross-Neveu quartic term [130–133], although one could also use a bosonic species to mimic an auxiliary field that carries the Gross-Neveu interactions [134]. For $D = 2 + 1$ dimensions, when working with an irreducible representation and two-component spinors, the Hubbard interaction maps again into a Gross-Neveu quartic term [95, 96]. In the current reducible case, where we have four-component spinors, the Hubbard-type interaction is richer and contains both inter-layer density-density interactions among all pairs of spinor states, as well as spin-changing collisions that involve effective inter-layer pair tunnellings (see Fig. 1 **(d)**).

## 3.2 Spin-3/2 atoms and 4-Fermi interactions

In this section, we present the details of how spin-3/2 fermionic atoms can naturally lead to the 4-Fermi interactions in Eq. (18). This is an example of the unique opportunity emphasised in the introduction: the possibility of tailoring local and global symmetries that connect to interesting models of high-energy physics. This brings us closer to the field of quantum simulators [34–37]; controllable quantum many-body systems that behave according to a specific model of interest [33]. In the context of quantum simulators for high-energy physics (see the reviews [135–143]), there have been several proof-of-principle experiments showing the quantum simulation of relativistic QFTs [18, 19, 144–177], including lattice gauge theories [121, 178]. The case of gauge theories is particularly demanding in terms of the required resources, as the tailored symmetries must be local and, ultimately, non-Abelian, requiring the introduction of additional gauge degrees to allow for this local symmetries. On the other hand, for synthetic Dirac matter with quartic interactions, the requirements are in principle milder, as one restricts to global and spacetime symmetries, including non-Abelian ones, but one can dispense with the extra gauge degrees of freedom.

Let us consider a gas of fermionic neutral atoms that can be tightly confined by optical potentials in a square lattice $\boldsymbol{x} = \sum_j \frac{\lambda_{\mathrm{L}}}{2}(n_j - N_j/2)\mathbf{e}_j$ with $\boldsymbol{n} \in \Lambda_s = \mathbb{Z}_{N_1} \times \mathbb{Z}_{N_2}$. We emphasise that the physical lattice spacing is set by half the wavelength $\lambda_{\mathrm{L}}/2$ of the laser that leads to the optical-lattice potential, which is kept fixed in the experiment. This physical lattice spacing will not be mapped onto the lattice-field-theory spacing $a$, which must be sent to $a \to 0$ to recover the continuum limit. Another difference is that, in second quantization [179–181], the atoms are described by dimensionless operators $f_{\boldsymbol{n},\sigma}^\dagger$ ($f_{\boldsymbol{n},\sigma}$) that create (annihilate) an atom in the position specified by $\boldsymbol{n}$, and in the internal electronic internal state given by $\sigma \in \mathcal{S}_\sigma$. The set $\mathcal{S}_\sigma$ generally depends on the particular type of atom, and its specific isotope, which can also control the bosonic/fermionic nature of the operators. We will be interested in the fermionic case, where

$$\left\{ f_{\boldsymbol{n}_1,\sigma_1}, f_{\boldsymbol{n}_2,\sigma_2}^\dagger \right\} = \delta_{\boldsymbol{n}_1,\boldsymbol{n}_2} \delta_{\sigma_1,\sigma_2}. \tag{19}$$

In the tight-binding regime, the system is thus described by a spin-conserving Hamiltonian, and its non-interacting part can be written [179–181] in second quantization as

$$H_{\mathrm{sc}} = \sum_{\boldsymbol{n}} \sum_{\sigma,j} \left( -\tilde{t}_j f_{\boldsymbol{n},\sigma}^\dagger f_{\boldsymbol{n}+\mathbf{e}_j,\sigma} + \mathrm{H.c.} \right), \tag{20}$$

where $\tilde{t}_j$ is the standard nearest-neighbour tunnelling coupling along the $j \in \{1, 2\}$ axis, and we set $\tilde{t}_1 = \tilde{t}_2 =: \tilde{t}$.

In order to find the set of internal states $\mathcal{S}_\sigma$, we need to consider the atomic energy level structure, focusing in particular in the groundstate manifold. For instance, in the case of $^6$Li Alkali atoms, we have principal quantum number $n = 2$, total orbital angular momentum $L = 0$ and spin $S = 1/2$ which, in spectroscopic notation leads to the $2^2S_{1/2}$ groundstate manifold. The total nuclear spin is $I = 1$, which leads to a couple of hyperfine levels with total angular momentum $F \in \{1/2, 3/2\}$. If we focus on the lower-energy state $F = 1/2$, the set of internal states is given by the two possible Zeeman sub-levels $M_F = \pm\frac{1}{2}$, namely $\mathcal{S}_\sigma = \{-\frac{1}{2}, +\frac{1}{2}\} =: \{0, 1\}$. At sufficiently cold temperatures, the scattering of the dilute Fermi gas is dominated by $s$-wave collisions between pairs of $^6$Li atoms, which mostly contribute [179] to a Hubbard density-density interactions that can be written as

$$H_{\text{int}} = \sum_{\boldsymbol{n}} \sum_{\sigma_1 \neq \sigma_2} \tfrac{1}{2} U_{\sigma_1 \sigma_2} n_{\boldsymbol{n}, \sigma_1} n_{\boldsymbol{n}, \sigma_2}, \tag{21}$$

where $n_{\boldsymbol{n}, \sigma} = f_{\boldsymbol{n}, \sigma}^\dagger f_{\boldsymbol{n}, \sigma}$ is the number operator, and $U_{\sigma_1 \sigma_2} = U_0 = \sqrt{8/\pi} k_L a_0 E_R \left( V_{0,x} V_{0,y} V_{0,z} / E_R^3 \right)^{1/4}$ is the Hubbard coupling strength [32]. Here, $k_L = 2\pi/\lambda_L$ is the laser wavevector, and $E_R = k_L^2/2m_a$ is the recoil energy of the $^6$Li atoms of mass $m_a$. In this expression, we have introduced the optical-potential depths along the different axes, which will be constrained to $V_{0,z} \gg V_{0,x}, V_{0,y}$ such that the dynamics takes place within the $xy$ plane. Finally, a key quantity in the Hubbard coupling is the s-wave scattering length $a_0$, which only depends on singlet scattering channel [32]. At cold temperatures and within the lowest hyperfine multiplet, the inter-atomic potential is rotationally invariant within the total angular momentum of the colliding pair $\boldsymbol{F}_t = \boldsymbol{F}_1 + \boldsymbol{F}_2$, which in this case leads to a couple of channels with $F_t \in \{0, 1\}$. Due to Pauli exclusion principle and the effective contact interactions for the $s$-wave channel, only the singlet case $F_t = 0$ is allowed, which is described by the above $s$-wave scattering length $a_0$. We note that the Hubbard interaction in Eq. (21) has a global SU(2) symmetry and, moreover, its strength can be controlled via Feshbach resonances by e.g. applying a magnetic field [182].

This type of interaction (21) would suffice to make connections to Gross-Neveu interactions for an irreducible representation of the gamma matrices [95, 96, 131], since the spinor components are only two. In this work, however, we are interested in reducible representations with a larger number spinor components $\mathcal{S}_\sigma = \{0, 1, 2, 3\}$, where a larger non-Abelian symmetry appears in the interactions. A well-known example of large non-Abelian global symmetries in the scattering appears for other atomic species, such as $^{87}$Sr Alkaline-earth atoms. In this case, there are two valence electrons, and the groundstate manifold has principal number $n = 5$, and vanishing total spin and orbital angular momentum $S = L = 0$, leading to the manifold $5^1S_0$. For vanishing $J = 0$, there is no hyperfine splitting due to the nuclear spin, such that $F = I = 9/2$, and we get a single multiplet with $N = 10$ Zeeman sub-levels $M_F \in \{-9/2, -7/2, \cdots, 9/2\}$. Since there is no hyperfine coupling, the atoms all interact with an $s$-wave scattering length that is independent of the nuclear features and, thus, equal for all of the $N = 10$ sub-levels $U_{\sigma_1 \sigma_2} = U_0 \ \forall \sigma_1 \neq \sigma_2$. Hence, Eq. (21) has an exact SU($N$) symmetry [183, 184]. When considering also the long-lived $5^3P_0$ level, one gets more flexibility, leading to the so-called two-orbital SU($N$) Hubbard models [185]. When considering a mixed-species Fermi gas with a couple of alkaline-earth atoms, the inter- and intra-orbital scattering preserve the SU($N$) symmetry. Provided that one can control their corresponding scattering lengths via Feshbach resonances, there are specific conditions where the two-orbital $SU(N)$ interactions would connect to the Gross-Neveu term between $N$ fermion flavours.

In this article, however, we are interested in a specific type of 4-Fermi term that goes beyond the Gross-Neveu couplings (18). These interactions, even for a single fermion flavor, have a non-Abelian SO(5) symmetry. As realised in [40, 41], it turns out that there is an exact SO(5)

symmetry in the theory of $s$-wave scattering when working with spin-3/2 alkali gases, similar to the case of $^6$Li for the $F = 3/2$ hyperfine multiplet. The only caveat is that we should consider other atomic species in which the $F = 3/2$ multiplet $\mathcal{S}_\sigma \in \{-3/2, -1/2, 1/2, 3/2\} =: \{0, 1, 2, 3\}$ corresponds to the lowest-energy level, as the scattering of the higher-energy hyperfine levels can otherwise lead to processes that bring the atoms into the lower hyperfine multiplet [186]. There are various possible Alkaline-earth atoms, such as the fermionic isotope $^{132}$Cs, which fulfill this condition and have an $F = 3/2$ low-energy multiplet. To the best of our knowledge, experiments with Cesium have been reported only for the bosonic $^{133}$Cs isotope, e.g. [187, 188]. Other possibilities would be to work with the atomic species $^9$Be, $^{135}$Ba, $^{137}$Ba, and $^{201}$Hg.

Following our discussion above, the total angular momentum of a colliding pair could be $F_t \in \{0, 1, 2, 3\}$ in this case, where Pauli exclusion principle forbids the odd-momentum channels. We thus have a pair of scattering lengths for the singlet $a_0$ and quintet $a_2$ channels. The contact interactions can be written in terms of projection operators on these two total angular momenta $P_{F_t}(\boldsymbol{n}) = \sum_{M_F} \sum_{\sigma_1, \sigma_2} \langle \sigma_1 \sigma_2 | F_t, M_{F_t} \rangle f_{\boldsymbol{n}, \sigma_1} f_{\boldsymbol{n}, \sigma_2}$ [186], namely

$$H_{\text{int}} = \sum_{\boldsymbol{n}} \sum_{F_t = 0,2} U_{F_t} P^\dagger_{F_t}(\boldsymbol{n}) P_{F_t}(\boldsymbol{n}), \tag{22}$$

which can be controlled by the individual coupling strengths

$$U_{F_t} = \sqrt{8/\pi} k_L a_{F_t} E_R \left( V_{0,x} V_{0,y} V_{0,z} / E_R^3 \right)^{1/4} .$$

As shown in [40], these interactions can be rewritten as a linear combination of 4-Fermi terms with a certain definition of gamma matrices. In order to connect to our previous discussion, we first need to define field operators with the right units, such that the spinor operator is $\Psi_{\boldsymbol{n}} = (f_{\boldsymbol{n},0}, f_{\boldsymbol{n},1}, f_{\boldsymbol{n},2}, f_{\boldsymbol{n},3})^t / a$, where $a$ is an effective lattice spacing that still needs to be connected to the microscopic cold-atom parameters. Using the corresponding Clebsch-Gordan coefficients $\langle \sigma_1 \sigma_2 | F_t, M_{F_t} \rangle$, and the specific $\alpha$-$\beta$ Dirac matrices (14)-(15), we find

$$H_{\text{int}} = a^2 \sum_{\boldsymbol{n} \in \Lambda_s} \frac{1}{2} \left( \tilde{g}^2 (\Psi^\dagger_{\boldsymbol{n}} \Psi_{\boldsymbol{n}})^2 - g^2 \left( (\Psi^\dagger_{\boldsymbol{n}} \boldsymbol{\alpha} \Psi_{\boldsymbol{n}})^2 + (\Psi^\dagger_{\boldsymbol{n}} \beta \Psi_{\boldsymbol{n}})^2 \right) \right), \tag{23}$$

where we have introduced the vector $\boldsymbol{\alpha} = (\alpha^1, \alpha^2, \alpha^3, \alpha^5)$, and the individual coupling constants

$$\frac{g^2}{a^2} = \frac{U_2 - U_0}{2}, \qquad \frac{\tilde{g}^2}{a^2} = \frac{3U_0 + 5U_2}{8} . \tag{24}$$

This brings us already really close to the desired 4-Fermi term in Eq. (18), which would require tuning $g^2 = \tilde{g}^2$, which would require setting $a_0 = -13a_2/11$, we would recover exactly the interaction term that combines Thirring, Gross-Neveu and squared mass terms. Let us emphasise, however, that the coupling proportional to $\tilde{g}^2$, which is proportional to the temporal component of the fermion current $J_0^2$, shall not play any role in the phase diagram of the fermionic QFT. Therefore, even if one cannot adjust the scattering lengths of the singlet and quintet channels, the physics will be completely equivalent, at least under the half-filling conditions explored in our work. In fact, as will also be clearer in our discussion below after discussing the cold-atom implementation of the quadratic part, there is a discrete SO(5) rotation corresponding to a 90° rotation, regardless of the relative value of these two couplings. This symmetry is responsible for protecting the higher-order topological state discussed in Sec. 4.1.

Figure 2: **Raman optical lattice: (a)** The atom cloud (light red sphere) is subjected to a 3D optical lattice, created using three pairs of counter-propagating laser beams (blue arrows) with mutually orthogonal polarizations (green arrows) and frequencies $\omega_S$. The atoms can be then confined into a 2D plane by increasing the potential depth in the $z$ direction. Additionally, travelling Raman beams (orange arrows) with frequencies $\omega_j^{\sigma\sigma'}$ and appropriate polarizations (red arrows) generate spin-changing processes, as discussed in the main text. Finally, a magnetic field $\mathbf{B}_{\text{ext}}$ is applied in the $z$ direction to split the hyperfine atomic energy levels, and a gradient obtained by lattice acceleration creates and energy difference $\Delta$ between nearest-neighbour sites in the $y$ direction. The panels **(b)**, **(c)** and **(d)** depict the two-photon Raman transitions giving rise to spin-changing processes, where $\varepsilon_\sigma$ is the electronic energy of $|\sigma\rangle$, and $\delta_j^{\sigma\sigma'}$ the corresponding detunings.

### 3.3 Raman lattices and twisted Wilson fermions

Let us now discuss how to realise the twisted Wilson mass regularization in the cold-atom system. In order to obtain the tunnelling structure required by Eq. (11), we need to extend Eq. (20) by considering additional Raman beams that also assist spin-changing tunnelling processes against certain energy offsets provided by the Zeeman effect of an external magnetic field. This experimental scheme falls within the so-called Raman optical lattices [159, 189–193], which have been exploited as quantum simulators of synthetic spin-orbit coupling [194–196]. In previous works [95, 96, 133, 197], we highlighted the potential of Raman optical lattices for the quantum simulation of Gross-Neveu-type QFTs with a standard Wilson discretization . For square lattices, these proposals connect to the recent realization of Chern insulators in [159]. The goal of this section is to present a Raman-lattice scheme for four spinor components that could serve as a quantum simulator of our SO(5) Dirac field theory regularised with the anisotropic twisted Wilson mass.

Let us first focus on tunnellings that change the atomic states corresponding to the spinor components $\mathcal{S}_\sigma \in \{0, 1, 2, 3\}$ by one unit, i.e. $\sigma \mapsto \sigma' = \sigma+1$, where we follow the conventions of Ref. [96]. Along the $x$ ($y$) axis, these tunnellings can be assisted by adding Raman beams along the $y$ ($x$) axis polarized in the $x$ ($z$) direction [Fig. 2**(a)**]. Due to the difference in polarization, the latter, together with the $z$ ($x$)-polarized standing wave responsible for the standard optical lattice along the $x$ ($y$) axis, give rise to two-photon spin-changing Raman processes. The key observation is that, due to the different spatial periodicity of the standard lattice and these Raman process, this spin-changing terms cannot contribute with on-site terms, but drive instead an assisted tunnelling. It can be shown that, in the tight-binding limit, this configuration gives rise to spin-changing tunnellings [189–192] that read

$$
\begin{aligned}
H_{\text{sf},j}^\sigma = -\sum_{\boldsymbol{n}} \Big[ & \mathrm{i} t_j^{\sigma\sigma+1} e^{\mathrm{i}(\delta_j^{\sigma\sigma+1} x^0 - \phi_{j,\boldsymbol{n}}^{\sigma\sigma+1})} f_{\boldsymbol{n},\sigma}^\dagger f_{\boldsymbol{n}+\mathbf{e}_j,\sigma+1} \\
& - \mathrm{i} t_j^{\sigma\sigma+1} e^{\mathrm{i}(\delta_j^{\sigma\sigma+1} x^0 - \phi_{j,\boldsymbol{n}}^{\sigma\sigma+1})} f_{\boldsymbol{n},\sigma}^\dagger f_{\boldsymbol{n}-\mathbf{e}_j,\sigma+1} + \text{H.c.} \Big],
\end{aligned}
\tag{25}
$$

where $t_j^{\sigma\sigma+1}$ is the corresponding strength of the Raman-assisted tunnelling along the $j$-th

axis, which must not be confused with the real time $x^0 = -i\tau$. Here, we have introduced $\phi_{j,n}^{\sigma\sigma+1} = \phi_j^{\sigma\sigma+1} - \pi(n_1 + n_2)$, where $\phi_j^{\sigma\sigma+1}$ is the relative phase between the standing wave and the Raman beam. We have also introduced $\delta_j^{\sigma\sigma+1} = \omega_S - \omega_j^{\sigma\sigma+1} - (\epsilon_\sigma - \epsilon_{\sigma+1})$, which is the corresponding detuning for the two-photon Raman transition, with $\omega_S$ and $\omega_j^{\sigma\sigma+1}$ the frequencies of the standing wave and the Raman beam, respectively, and $\epsilon_\sigma$ is the electronic energy for the level $\sigma$, which are controlled by the external magnetic field [see the two-photon transitions in Fig. 2**(b)** and **(c)**].

In order to realize the lattice field theory described by Eq. (11), we need to combine these spin-changing processes. In particular, we need to connect the states $(0,1)$ and $(2,3)$ both in the $x$ and $y$ directions, and choose the proper phases $\phi_j^{\sigma\sigma'}$, which can be checked by inspecting the structure of the $\mathbb{T}_1$ and $\mathbb{T}_2$ matrices in Eq. (12). Additionally, we need processes that flip the spinor twice in the $y$ direction, connecting the states $(0,2)$ and $(1,3)$. These can be obtained in a similar fashion by using instead $y$-polarized Raman beams in the $x$ direction [Fig. 2**(a)**], leading to the same expression as in Eq. (25) for the case $\sigma \mapsto \sigma' = \sigma+2$, where $\phi_2^{\sigma\sigma+2}$ is now the relative phase between two Raman beams and $\delta_2^{\sigma\sigma+2} = \omega_2^{\sigma+1\sigma+2} - \omega_2^{\sigma\sigma+1} - (\epsilon_\sigma - \epsilon_{\sigma+2})$, with $\omega_2^{\sigma+1\sigma+2}$ the frequency of the y-polarized Raman beam in the $x$ direction [Fig. 2**(d)**]. Let us now detail how the relative phases need to be tuned for each different tunnelling process.

We first focus on $\mathbb{T}_1$ (12), this is, the tunnelling processes along the $x$ axis. The corresponding spin-changing terms can be obtained by using two Raman beams with the same polarization, as explained above, connecting the pairs $(0,1)$ and $(2,3)$. The latter can be produced from the same laser source using acusto-optical modulators to generate beams with different detunings $\delta_j^{\sigma\sigma'}$ and phases $\phi_j^{\sigma\sigma'}$. Here, we choose in particular $\delta_x^{01} = -\delta_x^{23} =: \delta_x$ and $\phi_x^{01} = \phi_x^{23} = \pi$. After performing the following gauge transformation and rescaling

$$
\begin{aligned}
\Psi_{n,0} &= e^{i\frac{\delta_x}{2}x^0}\frac{f_{n,0}}{a}, & \Psi_{n,2} &= e^{-i\frac{\delta_x}{2}x^0}e^{i\pi(n_1+n_2)}\frac{f_{n,2}}{a}, \\
\Psi_{n,1} &= e^{-i\frac{\delta_x}{2}x^0}e^{i\pi(n_1+n_2)}\frac{f_{n,1}}{a}, & \Psi_{n,3} &= e^{i\frac{\delta_x}{2}x^0}\frac{f_{n,3}}{a},
\end{aligned}
\tag{26}
$$

it can be easily checked that the above configuration gives rise to the terms $a^2 \sum_n \left[\left(\Psi_n^\dagger \mathbb{T}_1 \Psi_{n+e_1} + \text{H.c.}\right) + \Psi_n^\dagger \mathbb{M}_1 \Psi_n\right]$ in Eq. (11), with the following parameters, $\tilde{m}_1 = \delta_x/2$, $t = 1/2a$, $r = t/\tilde{t}$, where we take $t := t_x^{01} = t_x^{23}$. We remark that the required $\mathbb{M}_1$ mass matrix (16) is generated by the Raman-assisted tunnelling once we move to the above rotating frame.

Let us now consider the spin-changing processes associated to $\mathbb{T}_2$ (12), for which we also add a spin-independent gradient along the $y$ axis,

$$
H_{\text{grad}} = \Delta \sum_{n_x,n_y} n_y \sum_\sigma f_{n,\sigma}^\dagger f_{n,\sigma},
\tag{27}
$$

which can be implemented e.g. by accelerating the optical lattice in that direction [198]. This gradient serves a two-fold purpose. First, for $\Delta \gg \tilde{t}$, it suppresses the spin-conserving tunnelling in the $y$ direction, which is absent in Eq. (11). Additionally, it allows us to tune the relative phase between the two terms in Eq. (25), as required by $\mathbb{T}_2$. Specifically, for each of the four spin-changing pairs of terms involved in $\mathbb{T}_2$, we employ two Raman beams, and choose the values of the detunings as follows: $\delta_y^{01} - \delta_x = \delta_y^{23} + \delta_x = -\tilde{\delta}_y^{01} + \delta_x = -\tilde{\delta}_y^{23} - \delta_x = \Delta$, where $\tilde{\delta}_y^{\sigma\sigma'}$ denotes the second Raman beam, as well as $\delta_y^{01} = \delta_y^{02}$, $\delta_y^{23} = \delta_y^{13}$, $\tilde{\delta}_y^{01} = \tilde{\delta}_y^{02}$ and $\tilde{\delta}_y^{23} = \tilde{\delta}_y^{13}$. For every pair of levels, this allows each Raman beam to independently assists one single spin-changing process in Eq. (25). The phases of these processes can now be chosen freely, which can be seen by transforming first to the interaction picture with respect to the gradient term in Eq. (27), and then applying the rotating-wave approximation

in the limit of large detunings. In particular, if we take $\phi_y^{01} = \tilde{\phi}_y^{01} = \phi_y^{23} = \tilde{\phi}_y^{23} = -\pi/2$ and $\phi_y^{02} = -\tilde{\phi}_y^{02} = -\phi_y^{13} = \tilde{\phi}_y^{13} = -\pi/2$, this configuration generates the terms $a^2 \sum_n \left( \Psi_n^\dagger \mathbb{T}_2 \Psi_{n+e_2} + \text{H.c.} \right)$ in Eq. (11) after applying again the transformations in Eq. (26), where we take $t_y^{01} = t_y^{23} = t$ and $t_y^{02} = t_y^{13} = \tilde{t}$. Finally, the mass term $a^2 \sum_n \Psi_n^\dagger \mathbb{M}_2 \Psi_n$ can be obtained by driving transitions between the $(0,1)$ and $(2,3)$ spinor pairs using microwave drivings with a Rabi frequency that gives the remaining microscopic parameter $\tilde{m}_2 = \Omega_y/2$.

Let us finally note that the relevant dimensionless parameter that appear in the phase diagrams to be discussed in the rest of the article correspond to

$$m_1 a = \frac{\delta_x}{4t} - r\,, \qquad m_2 a = \frac{\Omega_y}{4t} - r\,, \qquad \frac{g^2}{a} = \frac{U_2 - U_0}{4t}\,, \tag{28}$$

where we recall that the Wilson parameter is controlled by the ratio of the tunnellings $r = t/\tilde{t}$, and the fact that we have neglect the $\tilde{g}^2$ interaction as it will play no role (see the discussion below Eq. (24)). The important feature of this mapping is that all of the relevant parameters can be tuned independently in the experiments. As noted previously, the continuum limit does not require sending $\lambda_L \to 0$, but actually working in the vicinity of possible critical lines in parameter space $(m_1 a, m_2 a, g^2/a)$, which we start to explore below.

## 4 Higher-order topological insulators (HOTIs) under a discrete SO(5) rotation

In this section, we discuss the regions of parameter space where the free twisted-mass Wilson regularization can lead to HOTIs. In addition of presenting exact expressions for the zero-energy corner modes and the associated topological invariants, we will also discuss the symmetry responsible for protecting these topological states. Although the original SO(5) invariance of the 4-Fermi interactions is explicitly broken by the lattice discretization, we find a discrete SO(5) rotation that protects the groundstate topology. We finish this section with a discussion on how to detect these non-trivial topological properties in cold-atom experiments.

### 4.1 Flat bands and zero-energy corner modes

We now discuss the topological features of the half-filled groundstate in the absence of interactions $g^2 = 0$. By performing a Fourier transform $\Psi_n = \frac{1}{a\sqrt{N_s}} \sum_{k \in \text{BZ}} e^{ik \cdot na} \Psi_k$, the lattice model (11) becomes $H_0^{\text{TM}} = \sum_{k \in \text{BZ}} \Psi_k^\dagger \mathbb{H}_0(k) \Psi_k$, where the single-particle Hamiltonian can be written as

$$\mathbb{H}_0(k) \tag{29}$$
$$= \begin{pmatrix} -2t \sin(k_1 a)\sigma^x - 2t \sin(k_2 a)\sigma^y + (\tilde{m}_1 - 2\tilde{t}\cos(k_1 a))\sigma^z & (\tilde{m}_2 - 2\tilde{t}\cos(k_2 a))\mathbb{1}_2 \\ (\tilde{m}_2 - 2\tilde{t}\cos(k_2 a))\mathbb{1}_2 & 2t \sin(k_1 a)\sigma^x + 2t \sin(k_2 a)\sigma^y - (\tilde{m}_1 - 2\tilde{t}\cos(k_1 a))\sigma^z \end{pmatrix}.$$

Matrix diagonalization $\mathbb{H}_0(k)\left|\epsilon_{q,\pm}(k)\right\rangle = \epsilon_{q,\pm}(k)\left|\epsilon_{q,\pm}(k)\right\rangle$ with $q \in \{1,2\}$, yields a band structure with four energy bands $\epsilon_{q,\pm}(k) = \pm\epsilon(k)$ that display a two-fold degeneracy

$$\epsilon(k) = \sqrt{\sum_{j=1,2} \left( 4t^2 \sin^2(k_j a) + \left( \tilde{m}_j - 2\tilde{t}\cos(k_j a) \right)^2 \right)}\,. \tag{30}$$

This expression allows one to realise that, by setting $\tilde{m}_1 = \tilde{m}_2 = 0$ and $t = \tilde{t}$ (i.e. fixing $r = 1$ and $m_1 a = m_2 a = -1$), the energy bands become totally flat $\epsilon_{q,\pm}(k) = \pm 2t$. This is

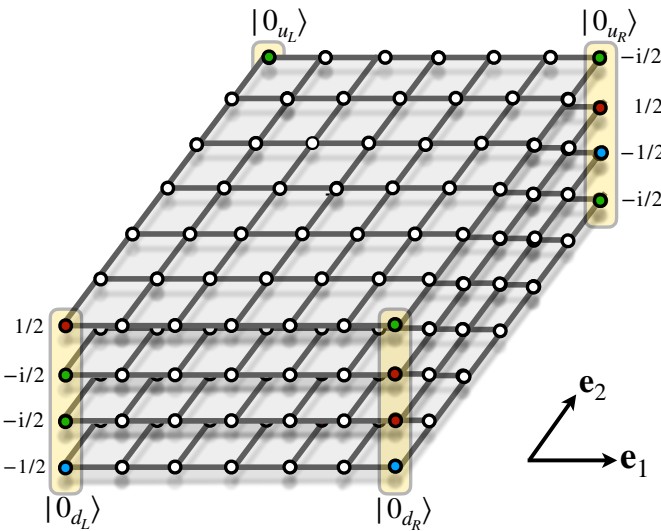

Figure 3: **Anomalous corner states in the flat-band regime:** For a finite multi-layer with parameters $m_1 = m_2 = r/a$ and $r = 1$, the Hamiltonian displays two-fold degenerate flat bands $\epsilon_{q,\pm}(\mathbf{k}) = \pm 2t$ and four zero-energy modes (31) that are strictly localised to the corners of the lattice. The colouring of the sites at the corners corresponds to the complex phases of Eq. (31).

the consequence of the Aharonov-Bohm phases depicted in Fig. 1, which lead to a destructive interference forbidding the tunnelling of bulk fermions more than two sites apart along any of the spatial directions (see, for instance, the vanishing amplitude of the two black-grey paths in Figs. 1 **(b)** and **(c)**). This can be understood as the phenomenon of Aharonov-Bohm caging [122], and finds its minimal manifestation at the corners of the multi-layer, where a single fermion with certain amplitudes over the various spinor components must remain localised. Considering that we have a total of $N_s = N_1 N_2$ lattice sites, with $N_j$ sites per spatial direction, we find that the states corresponding to such localised solutions on the corners $\mathbf{d}_L = (1,1)$, $\mathbf{d}_R = (N_1, 1)$, $\mathbf{u}_L = (1, N_2)$, and $\mathbf{u}_R = (N_1, N_2)$ have zero energy (see Fig. 3). These zero modes can be expressed as follows

$$
\begin{aligned}
\left|0_{\mathbf{d}_L}\right\rangle &= \tfrac{1}{2}\left(\Psi_{\mathbf{d}_L,0}^\dagger - i\Psi_{\mathbf{d}_L,1}^\dagger - i\Psi_{\mathbf{d}_L,2}^\dagger - \Psi_{\mathbf{d}_L,3}^\dagger\right)|\mathrm{vac}\rangle\,, \\
\left|0_{\mathbf{d}_R}\right\rangle &= \tfrac{1}{2}\left(\Psi_{\mathbf{d}_R,0}^\dagger + i\Psi_{\mathbf{d}_R,1}^\dagger + i\Psi_{\mathbf{d}_R,2}^\dagger - \Psi_{\mathbf{d}_R,3}^\dagger\right)|\mathrm{vac}\rangle\,, \\
\left|0_{\mathbf{u}_L}\right\rangle &= \tfrac{1}{2}\left(\Psi_{\mathbf{u}_L,0}^\dagger - i\Psi_{\mathbf{u}_L,1}^\dagger + i\Psi_{\mathbf{u}_L,2}^\dagger + \Psi_{\mathbf{u}_L,3}^\dagger\right)|\mathrm{vac}\rangle\,, \\
\left|0_{\mathbf{u}_R}\right\rangle &= \tfrac{1}{2}\left(\Psi_{\mathbf{u}_R,0}^\dagger + i\Psi_{\mathbf{u}_R,1}^\dagger - i\Psi_{\mathbf{u}_R,2}^\dagger + \Psi_{\mathbf{u}_R,3}^\dagger\right)|\mathrm{vac}\rangle\,.
\end{aligned}
\tag{31}
$$

As argued in the following section, these corner states are the boundary manifestation of certain topological invariants [199, 200] in the bulk bands of the system, which can lead to a topological quadrupole in analogy to the Bernevig-Benalcazar-Hughes (BBH) model [52, 53] of HOTIs, in particular to second-order topological insulators. When moving away from the flat-band limit, e.g. by increasing the masses $\tilde{m}_1 = \tilde{m}_2 > 0$ or by switching on the quartic interactions $g^2 > 0$, the perfect Aharonov-Bohm interference will disappear, and these zero modes will no longer be perfectly localised at the corners but, instead, start to spread within the bulk of the system. In particular, a certain localization length will emerge, which characterises the exponential decay of the corner-state amplitude as one moves towards the bulk. These corner states, which remain pinned to zero energy until the bulk energy gap is closed, are an example of anomalous boundary states, which only exist in the presence of the bulk. Moreover, they are protected by a certain discrete SO(5) rotation.

We previously argued that, in the long-wavelength limit around one of the Dirac points (7), the Euclidean action (8) recovers the SO(3) rotational symmetry of the Lorentz group. In the Hamiltonian formulation, this corresponds to the SO(1, 2) group of boosts, and two-dimensional spatial rotations of angle $\theta$ within the $xy$ plane. Let us emphasise that this is a property of the long-wavelength limit, since the lattice regularised action (8) is not invariant under arbitrary rotations $k_1 \mapsto k_1 \cos\theta - k_2 \sin\theta, k_2 \mapsto k_2 \cos\theta + k_1 \sin\theta$. On the other hand, there may exist other spatial symmetries that are exact for the full lattice model, including the 4-Fermi interactions (4). For instance, as discussed in more detail in Appendix B, there are two parity symmetries which, at the level of the fermionic field operators, act as follows

$$
\begin{aligned}
\Psi_{\boldsymbol{n}} &\mapsto \mathcal{P}_1 \Psi_{\boldsymbol{n}} = \gamma^1 \Psi_{(-n_1, n_2)}, \\
\Psi_{\boldsymbol{n}} &\mapsto \mathcal{P}_2 \Psi_{\boldsymbol{n}} = \gamma^2 \Psi_{(n_1, -n_2)}.
\end{aligned}
\tag{32}
$$

These transformations correspond to mirror symmetries that take either $(k_1, k_2) \mapsto (-k_1, k_2)$ or $(k_1, k_2) \mapsto (k_1, -k_2)$, and clearly commute with the single-particle Hamiltonian in Eq. (29): $\gamma^1 \mathbb{H}_0(-k_1, k_2)\gamma^1 = \gamma^2 \mathbb{H}_0(k_1, -k_2)\gamma^2 = \mathbb{H}_0(k_1, k_2)$. The composition of the two parities corresponds to the lattice inversion, which is a symmetry of the full Hamiltonian corresponding to a specific SO(1, 2) rotation with angle $\pi$.

We remark that the above symmetries need not exhaust all possibilities, as there may be additional spatial symmetries that are not connected to the SO(1, 2) group. The possibility of finding such discrete symmetries becomes clear when inspecting the form of the interacting term (18) which, in fact, allows for generic SO(5) rotations. In the Hamiltonian formulation, these admit a unitary representation, and do not include boosts but only spatial rotations. By simple inspection, it is clear that the first quartic-term $\sum_{\boldsymbol{n}} (\Psi_{\boldsymbol{n}}^\dagger \Psi_{\boldsymbol{n}})^2$ that appears in Eq. (18) will be a scalar under any such unitary transformation. In contrast, the remaining terms can be rewritten as $\sum_{\boldsymbol{n}} \boldsymbol{N}_{\boldsymbol{n}}^2$, which involves the norm of a 5-component vector of fermion bilinears

$$
\boldsymbol{N}_{\boldsymbol{n}} = \Psi_{\boldsymbol{n}}^\dagger \big(\beta, \alpha^1, \alpha^2, \alpha^3, \alpha^5\big)\Psi_{\boldsymbol{n}}.
\tag{33}
$$

Accordingly, this part of the 4-Fermi interaction is also invariant under $SO(5)$ rotations $\boldsymbol{N}_{\boldsymbol{n}} \mapsto R\boldsymbol{N}_{\boldsymbol{n}'}$, where $R$ is an orthogonal matrix $R^{\mathrm{t}}R = \mathbb{1}$ with $\det R = 1$, and $\boldsymbol{n}'$ is the corresponding two-dimensional rotation of the lattice.

Although the twisted-mass Wilson regularization (11) breaks explicitly this arbitrary SO(5) symmetry, there can exist specific rotation angles $\theta$ and matrices $R$ that correspond to an exact invariance of the model under a discrete SO(5) rotation. In particular, let us focus on a discrete $\pi/2$-rotation that transforms the spatial coordinates as $\boldsymbol{n} = (n^1, n^2) \mapsto \boldsymbol{n}' = (-n^2, n^1)$. It is important to emphasise that the action of this rotation on the Dirac spinor $\Psi_{\boldsymbol{n}} \mapsto S_R \Psi_{\boldsymbol{n}'}$ will *not* be the same as that of the corresponding SO(1, 2) Lorentz rotation $\Lambda$, namely $\Psi_{\boldsymbol{n}} \mapsto S_\Lambda \Psi_{\boldsymbol{n}'}$. As discussed in Appendix B, such a Lorentz rotation is $S_\Lambda = \exp\{\frac{1}{2}\theta\gamma^1\gamma^2\} = \mathbb{1}_2 \otimes \exp\{\frac{i}{4}\pi\sigma^z\}$, and one can then check that invariance of the lattice model $H_0^{\mathrm{TM}} \mapsto H_0^{\mathrm{TM}}$ would require a momentum-independent mass, which is no longer the case with the Wilson-mass regularisation (9). In order to find a $\pi/2$-rotation that leaves the model invariant we can, instead, look for a different rotation $S_R$ within the above SO(5) group of rotations. This SO(5) invariance requires that the Dirac $\alpha$ and $\beta$ matrices in Eqs. (14)-(15) transform as

$$
\begin{aligned}
S_R^\dagger \alpha^1 S_R &= +\alpha^2, & S_R^\dagger \alpha^2 S_R &= -\alpha^1, \\
S_R^\dagger \alpha^3 S_R &= +\alpha^5, & S_R^\dagger \alpha^5 S_R &= +\alpha^3, & S_R^\dagger \beta S_R &= -\beta.
\end{aligned}
\tag{34}
$$

We note that the first row of Eq. (34) coincides with the transformations that would be obtained from the $\pi/2$-rotation of the SO(1, 2) Lorentz group. On the other hand, the second row

describes different transformation laws that are crucial to attain invariance of the twisted Wilson mass (11). One can check that Eq. (34) has the following solution

$$\Psi_{\boldsymbol{n}} \mapsto S_R \Psi_{\boldsymbol{n}'}, \qquad S_R = \frac{1}{\sqrt{2}}\begin{pmatrix} S^\dagger & S \\ S & -S^\dagger \end{pmatrix}, \tag{35}$$

where $S = \exp\{\mathrm{i}\frac{\pi}{4}(1-\sigma^z)\}$ is known as the phase-gate in quantum computing [201], which maps the eigenvectors of $\sigma^x$ onto those of $\sigma^y$, namely $S\left|\pm_x\right\rangle = \left|\pm_y\right\rangle$. Combining these transformations with the action of the rotation on the crystal momenta, we find that the Bloch Hamiltonian (29) is indeed invariant when $m_1 = m_2$: $S_R\mathbb{H}_0(k_2,-k_1)S_R^\dagger = \mathbb{H}_0(k_1,k_2)$. At the level of the zero modes (31), the spatial part of this discrete SO(5) transformation respects the set of corners, and one says that these anomalous boundary states are protected by this discrete spatial symmetry. Since they only have support on a region of codimension 2, they are the boundary manifestation of symmetry-protected HOTI groundstates. We note that this symmetry can be interpreted as the multi-layer counterpart of the $\mathcal{C}_4$ symmetry of the BBH model [52,53].

Before closing this subsection, we remark that the unitary transformation on the spinors $\Psi_{\boldsymbol{n}} \mapsto S_R\Psi_{\boldsymbol{n}'}$, together with the transformation of the Dirac $\alpha$ and $\beta$ matrix (34), can be used to show that the vector of bilinears simply transforms as

$$\boldsymbol{N}_{\boldsymbol{n}} \mapsto \boldsymbol{N}_{\boldsymbol{n}'} = \Psi_{\boldsymbol{n}'}^\dagger\big(-\beta, \alpha^2, -\alpha^1, \alpha^5, \alpha^3\big)\Psi_{\boldsymbol{n}'}. \tag{36}$$

It is then clear that its norm is conserved and, thus, the quartic interactions (18) are also left invariant under this discrete SO(5) rotation $H_{\mathrm{int}} \mapsto H_{\mathrm{int}}$. Altogether, this proves that the full lattice model is invariant $H = H_0^{\mathrm{TM}} + H_{\mathrm{int}} \mapsto H$. Since the set of all four corners is also invariant under a $\pi/2$ rotation, we expect that the anomalous corner states will be protected by this symmetry and, thus, robust when varying the microscopic parameters of the model and switching on interactions. The only possibility to get rid of them is by closing the bulk energy gap, which would signal a quantum phase transition to a trivial band insulator or to a groundstate with symmetry-broken long-range order. We will explore these possibilities in the sections below but, first, let us provide a bulk perspective of the higher-order topology associated to these corner modes.

## 4.2 Higher-order topological invariants

According to our current understanding of the bulk-boundary correspondence of symmetry-protected topological phases [120], the anomalous zero modes are a boundary manifestation of a topological band structure in the bulk. For the present model, this phase should be a HOTI of second order with a certain non-vanishing topological invariant. As shown in [199, 200], this invariant can be expressed as the product of two winding numbers, which will allow us to find the phase transitions between the HOTI and a trivial insulator for $g^2 = 0$.

At the level of the twisted-mass free Hamiltonian (11), one sees that $\beta\mathbb{H}_0(\boldsymbol{k})\beta = -\mathbb{H}_0(\boldsymbol{k})$, which corresponds to the sublattice symmetry in the classification of topological insulators under non-spatial symmetries, and shows that the band structure (30) always comes in pairs of positive-negative energies. We note that single-particle Hamiltonian also has time-reversal and particle-hole non-spatial symmetries, such that it would belong to the BDI class. Rather than pursuing the standard manifestation of such a topological-insulator class, the idea now is to find an alternative unitarily-equivalent representation of the Dirac matrices that intertwines the effect of the sublattice symmetry with the dimensionality of the problem in a specific manner, which we now discuss. This is, once more, only allowed by the fact that we are working with a reducible representation of the Clifford algebra. By applying the following unitary

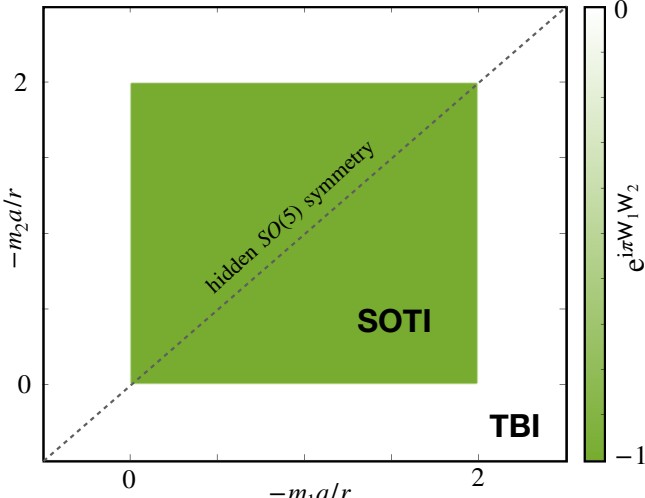

Figure 4: **Non-interacting higher-order topological phase diagram:** We represent the topological invariant in Eq. (45) as a function of the bare twisted masses $m_1$ and $m_2$. In the green inner square $m_j a \in (-2r, 0)$, the topological invariant is non-trivial $e^{i\pi W_1 W_2} = -1$, and the groundstate of the twisted-Wilson lattice model corresponds to a higher-order topological insulator (HOTI). The shaded line with $m_1 = m_2$ represents the regime where the model has a discrete SO(5) symmetry that protects the corner modes. The white region where $m_j a \notin (-2r, 0)$ for one or both masses corresponds to a trivial band insulator (TBI) with trivial topological invariant $e^{i\pi W_1 W_2} = +1$.

transformation $U$, built again from the $S$-gate, one finds that the $\beta$-matrix transforms as

$$U = \frac{1}{\sqrt{2}} \begin{pmatrix} \mathrm{i}S^\dagger & S \\ -\mathrm{i}S & S^\dagger \end{pmatrix}, \qquad U^\dagger \beta U = \sigma^y \otimes \sigma^z =: \beta_2 \otimes \beta_1, \tag{37}$$

where $\beta_1 = \sigma^z$ and $\beta_2 = \sigma^y$. On the other hand, the Dirac $\alpha$ matrices transform according to the following expressions

$$
\begin{aligned}
U^\dagger \alpha^1 U &= +\mathbb{1}_2 \otimes \sigma^x, & U^\dagger \alpha^2 U &= +\sigma^x \otimes \sigma^z, \\
U^\dagger \alpha^3 U &= -\mathbb{1}_2 \otimes \sigma^y, & U^\dagger \alpha^5 U &= -\sigma^z \otimes \sigma^z,
\end{aligned} \tag{38}
$$

which can be used to show that the transformed Bloch Hamiltonian (29) has the following tensor-product structure

$$U^\dagger \mathbb{H}_0(\boldsymbol{k}) U = \mathbb{1}_2 \otimes \hbar_1(k_1) + \hbar_2(k_2) \otimes \beta_1. \tag{39}$$

Here, we have defined the following single-particle Hamiltonians that only depend on the kinetic energy and Wilson mass along a specific direction in momentum space

$$
\begin{aligned}
\hbar_1(k_1) &= t \sin(k_1 a)\sigma^x - (\tilde{m}_1 - 2\tilde{t}\cos(k_1 a))\sigma^y, \\
\hbar_2(k_2) &= t \sin(k_2 a)\sigma^x - (\tilde{m}_2 - 2\tilde{t}\cos(k_2 a))\sigma^z.
\end{aligned} \tag{40}
$$

One can check that each of these Hamiltonians has an individual sublattice symmetry $\beta_1 \hbar_1(k_1)\beta_1 = -\hbar_1(k_1)$, and $\beta_2 \hbar_2(k_2)\beta_2 = -\hbar_2(k_2)$, which guarantees that the full Hamiltonian fulfills the desired transformation $\beta \mathbb{H}_0(\boldsymbol{k})\beta = -\mathbb{H}_0(\boldsymbol{k})$. Each of these terms (40) can

be understood as a two-band model corresponding to an un-twisted Wilson regularization of a $(1+1)$-dimensional Dirac-fermion QFT [131]. This tensor-product construction also highlights that the corner states are not equivalent to the edge states of topological insulators for one-dimensional chains, arranged along the boundaries of a square lattice. It is really the two-dimensional bulk that is required to host and protect these corner modes.

The lower-dimensional band structures of $\approx_j(k_j)$ have a pair of Dirac points corresponding to the projection of the previous Dirac points (7) onto the respective axis $k_{D,\ell_j} = \boldsymbol{k}_{D,\ell} \cdot \mathbf{e}_j$, each of which presents a different mass

$$k_{D,j} = \frac{\pi}{a}\ell_j, \qquad \ell_j \in \{0,1\}, \qquad m_j(\boldsymbol{k}_{D,\ell}) = \tilde{m}_j - 2\tilde{t}(-1)^{\ell_j}. \tag{41}$$

In analogy to our discussion of the Chern insulator (C.6) an the mass matrix (C.4), we can now define two mass matrices

$$M_{W,1} = \sum_{l_1=0,1} m_1(\boldsymbol{k}_{D,\ell})|\ell_1\rangle\langle\ell_1|, \qquad M_{W,2} = \sum_{l_2=0,1} m_2(\boldsymbol{k}_{D,\ell})|\ell_2\rangle\langle\ell_2|, \tag{42}$$

each of which contains the information about each of the twisted Wilson masses at the corresponding projection of the Dirac point. The Berry connection for each of these projections is $\mathcal{A}_i(k_i) = -\mathrm{i}\langle\epsilon_-(k_i)|\partial_{k_i}|\epsilon_-(k_i)\rangle$, where $|\epsilon_-(k_i)\rangle$ are the negative-energy modes that would be filled in the corresponding lower-dimensional groundstate. One can define a Chern-Simons form [82] associated to this Berry connection or, equivalently, a so-called Zak's phase [202], which plays the role of the above Chern number (C.6) in this reduced dimensionality. We find that each of these invariants

$$\mathsf{CS}_j = \frac{1}{2\pi}\int \mathrm{d}k_j \mathcal{A}_j(k_j) = \frac{1}{2\pi}\arg\left\{\mathrm{Det}(M_{W,j})\right\}, \tag{43}$$

is again non-trivial when an odd number of the projected Dirac points have a negative twisted Wilson mass

$$\mathsf{CS}_j = \frac{1}{4}\sum_{\ell_j=0,1}(-1)^{\ell_j}\mathrm{sgn}\{m_j(\boldsymbol{k}_{D,\ell})\}. \tag{44}$$

As is well-known for standard first-order topological insulators [203], these topological invariants $\mathsf{CS}_j = \mathsf{W}_j/2$ are proportional to the winding number $\mathsf{W}_j$ of the mappings $\hat{\boldsymbol{d}}_j(k_j) = \boldsymbol{d}_j(k_j)/||\boldsymbol{d}_j(k_j)|| : \mathrm{BZ} \mapsto U(1)$, where $\boldsymbol{d}_j(k_j)$ is the vector of coefficients of the individual Hamiltonians (40) in the Pauli basis $h_j(k_j) = \boldsymbol{d}(k_j)\cdot\boldsymbol{\sigma}$. As discussed in Reference [82], one can define a topological invariant that is also gauge invariant by considering the Wilson loop associated to such winding number. In this way, by simply multiplying the winding numbers together and exponentiating them, we obtain a topological invariant for HOTIs with sublattice symmetry symmetry (39), which reads

$$\mathrm{e}^{\mathrm{i}\pi\mathsf{W}_1\mathsf{W}_2} = \begin{cases} -1, & \text{if } -2r < m_j a < 0, \qquad \forall j \in \{1,2\}, \\ +1, & \text{else.} \end{cases} \tag{45}$$

A non-vanishing invariant $\mathrm{e}^{\mathrm{i}\pi\mathsf{W}_1\mathsf{W}_2} = -1$ signals the non-trivial topology of the bulk, and must have a boundary manifestation in the form of corner states. Indeed, for $m_1 a = m_2 a = r$, which corresponds to the previous flat-band limit $\tilde{m}_1 = \tilde{m}_2 = 0$ when $r = 1$, we find that $\mathrm{e}^{\mathrm{i}\pi\mathsf{W}_1\mathsf{W}_2} = -1$ in the bulk, while the zero-energy corner states are those of Eq. (31). Away from this flat-band limit, and while $m_j a \in (-2r, 0)$ in both directions, the groundstate is still a HOTI with $\mathrm{e}^{\mathrm{i}\pi\mathsf{W}_1\mathsf{W}_2} = -1$, and the anomalous boundary states remain exponentially localised to the corners. The non-interacting HOTI in parameter space corresponds to the square displayed in Fig. 4. For mass parameters in the inner square, the groundstate is a HOTI whereas, outside this region, it is trivial.

In the next section, we will explore the fate of this HOTI as the fermion self-interactions $g^2$ are increased. Let us note that, in our $D = 3$-dimensional spacetime, considering the role of the quartic interactions by simple power counting can be misleading. Indeed, the dimensions of the Dirac field is $[\Psi_n] = \mathsf{L}^{-1}$, such that $[t_j] = [\tilde{t}_j] = [m_1] = [m_2] = \mathsf{L}^{-1}$, whereas the coupling strength has units of length $[g^2] = \mathsf{L}$ and, thus, a negative energy dimension. Naive power-counting arguments would then suggest the 4-Fermi interaction is perturbatively *irrelevant*, such that the QFT in $D = 2 + 1$ would be non-renormalizable. On the other hand, it is well-known that the relevant/irrelevant nature of the couplings can be modified after resummation in the large-$N$ limit [204], whereupon the Thirring-like interaction $(\Psi_n^\dagger \Psi_n)^2 + \sum_{j=1,2}(\Psi_n^\dagger \alpha^j \Psi_n)^2$ becomes *marginal*, whereas the Gross-Neveu-like interactions $(\Psi_n^\dagger \beta \Psi_n)^2$ become *relevant*. Even if the corresponding 4-Fermi field theories are not perturbatively renormalizable, they become $1/N$ renormalizable [8,25–28]. The goal of the following section is to analyse the effect of the 4-Fermi interactions on the HOTI, including also the mass-squared interactions $\sum_{j=3,5}(\Psi_n^\dagger \alpha^j \Psi_n)^2$, using the large-$N$ expansion.

### 4.3  Experimental detection

We finish this section by commenting different strategies to detect the non-trivial topological properties described above in ultracold atom experiments. First, localized boundary modes, such as corner modes, can be directly detected in real space by locally resolving the atomic density for different spin components using a quantum gas microscope [205,206]. This can be achieved thanks to the possibility to modify the optical potentials and to create sharp boundaries that localize the modes, resulting in an excess of atomic density at the boundaries for the topological phase as compared to the trivial phase, as recently demonstrated in 1D for non-interacting [207] as well as for an interacting Fermi-Hubbard cold-atom system [208]. These techniques can be applied also in 2D [62], where the excess density at the boundaries can be fitted to an exponential function, extracting the localization length that remains finite within the topological phase and diverges at the phase transition. We note that the presence of corner states can additionally be detected by measuring the quantized charge transport that follows an adiabatic Thouless pump [70]. Regarding the non-trivial bulk topology, this can be detected by inspecting the structure of the entanglement spectrum associated to certain bipartitions of the sytem [209], which can be extracted experimentally in atomic systems by measuring local densities and currents [210].

## 5  Correlated HOTIs, fermion condensates and quantum phase transitions

In this subsection, we discuss the effect of the 4-Fermi interactions in detail. We argued previously that the HOTI groundstate is protected by a discrete SO(5) rotation and, thus, should be robust under symmetric perturbations unless those are sufficiently strong such that the bulk energy gap closes allowing for a change of the topological invariant, or if a certain symmetry-breaking phase transition takes place. We also showed in Eq. (36) that the 4-Fermi interactions preserve this discrete SO(5) rotational symmetry, such that one expects the HOTI phase to become a correlated HOTI as one increases the coupling strength $g^2 > 0$. Eventually, when the interactions are sufficiently strong, there may be a symmetry-breaking phase transition at some $g_c^2$, which paves the way for the appearance of new phases of matter typically referred to as fermion condensates in the QFT literature. The goal of this section is to explore the possible condensates allowed by the rich SO(5) structure of the self-interactions, and provide a quantitative account about which of the fermion condensates is expected to form at which point in parameter space $(m_1 a, m_2 a, g^2/a)$.

## 5.1 Auxiliary fields and the discrete SO(5) rotation

To accomplish this goal, we shall return to the Euclidean formulation of our SO(5) Dirac matter in Eqs. (1) and (4), where we can make use of controlled approximations such as the large-$N$ expansion [204]. To present a non-perturbative, yet tractable, account of the 4-Fermi interactions, one can generalise the Euclidean action in Eqs. (1) and (4) to $N$ flavours

$$
\begin{aligned}
\psi(x) &\mapsto \psi(x) = (\psi_1(x), \psi_2(x), \cdots, \psi_N(x))^{\mathrm{t}}, \\
\overline{\psi}(x) &\mapsto \overline{\psi}(x) = (\overline{\psi}_1(x), \overline{\psi}_2(x), \cdots, \overline{\psi}_N(x)).
\end{aligned}
\tag{46}
$$

The free part of the regularised Wilson twisted-mass action $S_0^{\mathrm{TM}}$ (8) simply becomes a sum of the corresponding actions for each of the fermion flavours. In contrast, the 4-Fermi term does couple the different flavours where, to have a consistent $N \to \infty$ limit, one must rescale the coupling strength as

$$
S_{\mathrm{int}} = \int \mathrm{d}^3 x \, \frac{g^2}{2N} \Big( -J_\mu J^\mu + (\overline{\psi}\gamma^3\psi)^2 + (\overline{\psi}\gamma^5\psi)^2 - (\overline{\psi}\psi)^2 \Big),
\tag{47}
$$

where the gamma matrices for $N$ flavours appearing the in the above bilinears should be understood as $\gamma^a \mapsto \mathbb{1}_N \otimes \gamma^a$. We will assume this in all the expressions below, which leads to an additional $U(N)$ symmetry in flavour space.

The first step of the large-$N$ approximation is to introduce auxiliary fields via a Hubbard-Stratonovich transformation [211,212], such that the partition function becomes quadratic in the Grassmann spinors. We need to introduce six real bosonic fields $a_\mu(x), \sigma_1(x), \sigma_2(x),$ and $\pi(x)$, such that the partition function can be exactly rewritten as

$$
Z = \int [\mathrm{D}\overline{\psi}\mathrm{D}\psi \mathrm{D}a_\mu \mathrm{D}\sigma_j \mathrm{D}\pi] \mathrm{e}^{-S_0^{\mathrm{TM}} - S_{\mathrm{int}}' - \int \mathrm{d}^3 x \frac{N}{2g^2}\left(a_\mu a^\mu + \sigma_j \sigma^j + \pi^2\right)}.
\tag{48}
$$

As a consequence of this transformation, we get the following action coupling between the fermionic and bosonic fields

$$
S_{\mathrm{int}}' = \int \mathrm{d}^3 x \, \overline{\psi}(x) \big( \mathrm{i}\gamma^\mu a_\mu(x) + \mathrm{i}\gamma^3 \sigma_1(x) + \mathrm{i}\gamma^5 \sigma_2(x) + \pi(x) \big) \psi(x).
\tag{49}
$$

Except for the $\pi$-field, the rest can be incorporated in the free action $S_0^{\mathrm{TM}}$ by the following substitution

$$
\partial_\mu \mapsto \partial_\mu + \mathrm{i}a_\mu(x), \qquad m_j \mapsto m_j + \mathrm{i}\sigma_j(x),
\tag{50}
$$

which shows that the auxiliary fields $a_\mu(x)$ act as an effective gauge-like potential, the components of which admix by the Lorentz transformations, and couple to the fermions minimally. We stress, however, that the free action (48) of these auxiliary fields is not gauge invariant, but simply a mass-like term that becomes very heavy in the large-$N$ limit. On the other hand, the $\sigma_j(x)$ are scalar fields that couple to the fermions via a pair of twisted Yukawa-type couplings. To be more accurate, we should consider the Euclidean formulation of the two parity transformations (32), which amount to

$$
\begin{aligned}
\mathcal{P}_1 \psi(x) &= \gamma^1 \psi(\tau, -x_1, x_2), & \mathcal{P}_1 \overline{\psi}(x) &= -\overline{\psi}(\tau, -x_1, x_2)\gamma^1, \\
\mathcal{P}_2 \psi(x) &= \gamma^2 \psi(\tau, x_1, -x_2), & \mathcal{P}_2 \overline{\psi}(x) &= -\overline{\psi}(\tau, x_1, -x_2)\gamma^2.
\end{aligned}
\tag{51}
$$

Since the 4-Fermi interactions are invariant under these parity transformations, we know that $S_{\mathrm{int}}' \mapsto S_{\mathrm{int}}'$, which require the auxiliary gauge-like fields to transform as

$$
\begin{aligned}
\mathcal{P}_1 a_\mu(x) &= -(1 - 2\delta_{\mu,0}) a_\mu(\tau, -x_1, x_2), \\
\mathcal{P}_2 a_\mu(x) &= -(1 - 2\delta_{\mu,0}) a_\mu(\tau, x_1, -x_2),
\end{aligned}
\tag{52}
$$

whereas the auxiliary $\sigma$ fields transform as

$$
\begin{aligned}
\mathcal{P}_1 \sigma_j(x) &= \sigma_j(\tau, -x_1, x_2), \\
\mathcal{P}_2 \sigma_j(x) &= \sigma_j(\tau, x_1, -x_2).
\end{aligned}
\tag{53}
$$

Therefore, we see that the $\sigma$ fields are all parity even, the $a_0(x)$ component of the gauge-like field is parity even, while the spatial components $\boldsymbol{a}(x)$ are parity odd.

We have so far left out the discussion of the $\pi(x)$ field, as it gives rise to a new term that was not present in $S_0^{\mathrm{TM}}$. Considering the parity transformations in Eq. (51), we see that in order to recover parity invariance, the $\pi(x)$ field should transform as

$$
\begin{aligned}
\mathcal{P}_1 \pi(x) &\mapsto -\pi(\tau, -x_1, x_2), \\
\mathcal{P}_2 \pi(x) &\mapsto -\pi(\tau, x_1, -x_2),
\end{aligned}
\tag{54}
$$

and, thus, corresponds to a pseudo-scalar field that is parity odd. According to Eq. (49), this pseudo-scalar auxiliary field couples to the fermions via the standard Yukawa coupling.

Let us finally discuss the discrete SO(5) rotational symmetry (35) which, for the Euclidean Grassmann fields, corresponds to

$$
\psi(\tau, \boldsymbol{x}) \mapsto S_R \psi(\tau, x_2, -x_1), \qquad \overline{\psi}(\tau, \boldsymbol{x}) \mapsto -\overline{\psi}(\tau, x_2, -x_1) S_R^\dagger,
\tag{55}
$$

where we recall that $S_R$ is the unitary matrix given by Eq. (35). Considering that the Euclidean gamma matrices transform as

$$
\begin{aligned}
S_R^\dagger \gamma^1 S_R &= -\gamma^2, & S_R^\dagger \gamma^2 S_R &= +\gamma^1, \\
S_R^\dagger \gamma^3 S_R &= -\gamma^5, & S_R^\dagger \gamma^5 S_R &= -\gamma^3, & S_R^\dagger \gamma^0 S_R &= -\gamma^0,
\end{aligned}
\tag{56}
$$

one can readily see that the original Euclidean action Eqs. (1) and (4) is also invariant under this discrete SO(5) rotation (55) when $m_1 = m_2$. Once the auxiliary fields are introduced, this rotational symmetry implies that the vector field should transform as

$$
\begin{aligned}
a_0(x) &\mapsto +a_0(\tau, x_2, -x_1), \\
a_1(x) &\mapsto +a_2(\tau, x_2, -x_1), \\
a_2(x) &\mapsto -a_1(\tau, x_2, -x_1),
\end{aligned}
\tag{57}
$$

whereas the scalar and pseudo-scalar fields must fulfill

$$
\begin{aligned}
\sigma_1(x) &\mapsto +\sigma_2(\tau, x_2, -x_1), \\
\sigma_2(x) &\mapsto +\sigma_1(\tau, x_2, -x_1), \\
\pi(x) &\mapsto -\pi(\tau, x_2, -x_1).
\end{aligned}
\tag{58}
$$

One can check that these pair of equations (57) and (58) define a transformation on a vector of auxiliary fields $\boldsymbol{\phi}(x) := (a_0(x), a_1(x), a_2(x), \sigma_1(x), \sigma_2(x), \pi(x)) \mapsto O\boldsymbol{\phi}(\tau, x_2, -x_1)$, with $O^{\mathrm{t}}O = \mathbb{1}$ and $\det O = +1$. We could then say that, within the Euclidean formulation where the field and the adjoint are independent Grassmann fields, the rotational symmetry (55) can be interpret as a specific SO(6) rotation of the auxiliary fields and fermion bilinears. We note that the interacting part of the Euclidean action has indeed a larger symmetry under arbitrary $SO(6)$ rotations when expressed in term of auxiliary fields, which gets broken down by the lattice regularization of the free fermions. It is only for the specific $\pi/2$ rotation above that one recovers invariance of the full Euclidean action. However, it must be noted that this transformation is a rotation of the auxiliary fields about the axis $a_0(x) \mapsto a_0(x')$. If one considers the adjoint definition in the Hamiltonian approach, $a_0(x)$ should always remain invariant, such that the discrete symmetry reduces to the previous SO(5) rotation.

The idea of the large-$N$ method in the present context is to assume that these scalar fields will be homogeneous in the groundstate of the interacting theory $\boldsymbol{\phi}(x) = \boldsymbol{\Phi}$, $\forall x$, and try to determine the regime in parameter space $(m_1 a, m_2 a, g^2/a)$ where some of them achieve a non-zero vacuum expectation value. This is the region of parameter space where the groundstate supports a specific combination of fermion condensates

$$A_\mu = \langle a_\mu(x) \rangle, \qquad \Sigma_j = \langle \sigma_j(x) \rangle, \qquad \Pi = \langle \pi(x) \rangle. \qquad (59)$$

Each of these fermion condensates is responsible for breaking a particular symmetry, and may even change completely the QFT that governs the continuum limit in the vicinity of such a symmetry-breaking phase transition. This is the case of the vector condensate, which is proportional to a fermion current $A_\mu \propto \langle J_\mu \rangle$, and would thus forbid recovering Lorentz invariance even in the continuum limit. Such condensates have been identified in related two-band models [95, 96].

The vacuum expectation values of the scalar and pseudo-scalar condensates play a different role. In fact, the scalar condensates $\Sigma_1 \propto \langle \overline{\psi} i \gamma^3 \psi \rangle$ and $\Sigma_2 \propto \langle \overline{\psi} i \gamma^5 \psi \rangle$ are generally non-zero except for a particular line in parameter space $m_1 a = m_2 a = -r$, which is a consequence of the twisted Wilson mass regularization. These scalar condensates do not break any of the parity symmetries (32), but contribute with a renormalization of the bare twisted masses $m_j \mapsto m_j + \Sigma_j$ which, as will be discussed below, can change abruptly the value of the topological invariant (45). When the bare masses $m_1 = m_2 =: m$, the two scalar condensates take equal values $\Sigma_1 = \Sigma_2 =: \Sigma$, such that the discrete SO(5) protecting symmetry (58) is not broken, and one can still talk about the symmetry-protected HOTI. By increasing the interaction $g^2$, as discussed below, one of the possibilities is that the values of $\Sigma$ will change and lead to an interaction-induced quantum phase transitions between the correlated HOTI, and a trivial band insulator with no corner states and a vanishing many-body topological invariant.

Before closing this section, we comment on the remaining fermion condensate $\Pi \propto \langle \overline{\psi} \psi \rangle$. Although in even spacetime dimensions, this condensate is parity even and associated to the breakdown of chiral symmetry, in our even-dimensional QFT it plays a different role. As discussed above, this condensate is odd under any of the parities (32), and a finite vacuum expectation value would imply the spontaneous breakdown of parity. The possibility of finding such condensates in the standard Wilsonian lattice regularization of Dirac QFTs was initially considered by S. Aoki [213], and it is typically referred to as an Aoki condensate in lattice gauge theories. In our current model of HOTIs, rather than parity breaking, it is more important to consider the discrete SO(5) symmetry, which is spontaneously broken by a non-zero value of this $\pi$ condensate (58). We note that the formation of vector condensates $A \neq 0$ would also break spontaneously the protecting symmetry in light of Eq. (58). As shown below, understanding the competition of the different condensation channels is the key to understand the phase diagram of our HOTI.

## 5.2 Large-$N$ condensates and the effective potential

In this subsection, we describe the results of the aforementioned large-$N$ technique to chart the phase diagram of the interacting HOTI. As advanced previously, there are various possible fermion condensates characterised by different vacuum expectation values, which could be obtained in the $N \to \infty$ limit by solving a set of gap equations. As discussed in the context of lattice gauge theories [214], these gap equations are, however, only valid for non-vanishing values of the vacuum expectation values $\Phi_a \neq 0$. On the other hand, we are also interested in the competition of the HOTI with a trivial band insulator, where the symmetry-breaking fermion condensates vanish and there is no spontaneous symmetry breaking. In order to explore the whole phase diagram, we need to go beyond the gap equations and obtain the large-$N$

effective potential $V_{\text{eff}}(\mathbf{\Phi})$, the minimum of which will provide the values of the auxiliary fields in $\mathbf{\Phi}$ for any specific point in parameter space determining, in particular, which of the possible symmetry-breaking condensates prevail.

The large-$N$ effective potential can be obtained diagrammatically by considering the Feynman diagrams with a single fermion loop and an increasing number of external lines describing the auxiliary fields. Any other one-particle irreducible diagram with more fermion loops and internal auxiliary lines contributes with a higher order in $1/N$, and can thus be neglected when $N \to \infty$. In the standard calculation of the chiral-invariant Gross-Neveu QFT, one can show that only diagrams with an even number of external auxiliary legs can give a non-zero contribution [204]. On the other hand, for our twisted Wilson mass regularization, one cannot apply the same arguments, and must also take into account the diagrams with an odd number of external legs. A similar situation can also be found in Chern-insulator models with a standard Wilson discretization although, there, one finds that these odd terms are zero due to the vanishing of the corresponding integrals [95, 96]. For the present model, this is not the case, and both the even and odd Feynman diagrams have a non-zero contribution that must be accounted for (see Appendix D).

The resummation of these Feynman diagrams yields the leading-order quantum radiative corrections $\delta V_{\text{eff}}(\mathbf{\Phi})$ to the classical potential of the auxiliary fields

$$V_{\text{eff}}(\mathbf{\Phi}) = \frac{N}{2g^2}\mathbf{\Phi}^2 + \delta V_{\text{eff}}(\mathbf{\Phi}). \tag{60}$$

As we have assumed that these auxiliary fields are homogeneous, all the relevant information is included in this effective potential, which plays a crucial role in determining the parameter regimes where the groundstate displays non-zero vacuum expectation values $\mathbf{\Phi} \neq 0$. In fact, the classical part of the potential predicts a zero vacuum expectation value $\mathbf{\Phi} = 0$, and it is the contribution of radiative corrections $\delta V_{\text{eff}}(\mathbf{\Phi})$ for $g^2 > 0$ that can change the minimum of Eq. (60) allowing for condensation. As discussed in Appendix D, this resummation can be accomplished to all orders of $g^2$, which allows us to address non-perturbative effects in the phase diagram of the model. We find that the quantum correction can be expressed as

$$\frac{\delta V_{\text{eff}}}{2N} = -\int_k \log\left(\frac{k_0^2 + (\hat{\mathbf{k}} + \mathbf{A})^2 + \mathbf{m}^2(\mathbf{k}, \mathbf{\Sigma}) + \Pi^2}{k_0^2 + \hat{\mathbf{k}}^2 + \mathbf{m}^2(\mathbf{k}, \mathbf{0})}\right), \tag{61}$$

where we recall that $\hat{\mathbf{k}}$ is the regularised spatial momentum in Eq. (6), and we have introduced the shifts in the twisted Wilson masses (9) stemming from the additive renormalisations that one finds for a non-zero values by the scalar condensates

$$\mathbf{m}(\mathbf{k}, \mathbf{\Sigma}) = \left(m_1(\mathbf{k}) + \Sigma_1, m_2(\mathbf{k}) + \Sigma_2\right). \tag{62}$$

We also recall that the integral symbol is a short-hand notation (5) for the spatial mode sum and the zero-temperature limit of the Matsubara sum. As discussed in Appendix D, the temporal component of the pseudo-vector field $A_0$ does not contribute to the effective potential and, thus, cannot condense. The occurrence of other condensation channels will depend on the minimum of $V_{\text{eff}}(\mathbf{A}, \mathbf{\Sigma}, \Pi)$. For $m_1 = m_2$, this effective potential can be easily seen to be invariant under the discrete SO(5) rotation that takes $k_1 \mapsto k_2$, $k_2 \mapsto -k_1$, together with $A_1 \mapsto A_2, A_2 \mapsto -A_1, \Sigma_1 \mapsto \Sigma_2, \Sigma_2 \mapsto \Sigma_1$, and $\Pi \mapsto -\Pi$.

Let us note that the above expressions with the specific lattice regularization are in fact more general, and would also apply to continuum QFTs with the same 4-Fermi interactions. This would simply require substituting $\hat{\mathbf{k}} \mapsto \mathbf{k}$ and $m_j(\mathbf{k}) \mapsto m_j$, recovering in this way the underlying Lorentz invariance. On the other hand, the expressions could also be used with a discretized Euclidean time by substituting $k_0 \mapsto \sin(k_0 a)/a$ with $k_0 = -\pi/a + 2\pi(n_0 + 1/2)/aN_0$

with $n_0 \in \mathbb{Z}_{N_0}$. From the perspective of the cold-atom quantum simulator, one is interested in the continuum-time limit and the Hamiltonian field theory. In this case, one can actually perform the integral over $k_0 \in \mathbb{R}$, and express the radiative corrections as follows

$$\delta V_{\text{eff}}(\boldsymbol{\Phi}) = 2N \int_{\boldsymbol{k}} \left( \epsilon(\hat{\boldsymbol{k}} + \boldsymbol{A}, m(\boldsymbol{k}, \boldsymbol{\Sigma}), \Pi) - \epsilon(\hat{\boldsymbol{k}}, m(\boldsymbol{k}, \boldsymbol{0}), 0) \right), \tag{63}$$

where we have introduced a short-hand for the spatial-momenta sum within the Brillouin zone $\int_{\boldsymbol{k}} = \frac{1}{(N_s a)^2} \sum_{\boldsymbol{k} \in \text{BZ}}$. In addition, we have generalised the energy dispersion relation in Eq. (30) to account for the possible non-zero vacuum expectation values of the fermion condensates

$$\epsilon(\hat{\boldsymbol{k}} + \boldsymbol{A}, m(\boldsymbol{k}, \boldsymbol{\Sigma}), \Pi) = \sqrt{(\hat{\boldsymbol{k}} + \boldsymbol{A})^2 + m^2(\boldsymbol{k}, \boldsymbol{\Sigma}) + \Pi^2}. \tag{64}$$

Expression (63) has a very simple interpretation, the large-$N$ radiative corrections are given by the total shift of single-particle energy levels in the filled bands, i.e. those with negative energies forming the Dirac sea, once some of the condensates form $\boldsymbol{\Phi} \neq \boldsymbol{0}$. For the scalar condensates $\Sigma_j$, such corrections appear as soon as the interactions are switched on $g^2 > 0$. The only exception is the straight line at $m_1 a = m_2 a = -r$, in which the scalar condensates vanish by symmetry arguments. For the vector $\boldsymbol{A}$ and pseudo-scalar $\Pi$ condensates, the situation is completely different. A non-zero value of these condensates would break the discrete SO(5) symmetry in Eqs. (57)-(58), which can only happen spontaneously for a sufficiently-strong coupling $g^2 > g_c^2(m_1, m_2)$. In order to find out which of the condensates prevails, we need to minimize the full effective potential

$$\boldsymbol{\Phi}^\star = \text{argmin} \left\{ \frac{N}{2g^2} \boldsymbol{\Phi}^2 + \delta V_{\text{eff}}(\boldsymbol{\Phi}) \right\}, \tag{65}$$

which is an unconstrained multi-parameter non-linear minimization problem that must be addressed numerically, as we detail in the following subsection.

## 5.3 Self-energy and correlated HOTIs

In the previous subsection, we have discussed the procedure to find the values of $\boldsymbol{\Phi}^\star$, which will allow us to detect the symmetry-breaking condensates and localise the critical lines that separate them from the HOTI and the trivial band insulator. On the other hand, we would also like to predict the flow of the critical lines separating the HOTI from the trivial band insulator as the coupling increases $g^2 > 0$. Since topological phases cannot be distinguished by a local order parameter, we also need to calculate the topological invariant (45) away from the non-interacting free theory. An approximation that has already been used for the many-body topological invariants of standard topological insulators [215–219], deals with the so-called topological Hamiltonian.

Many-body topological invariants can be defined via the two-point Green's functions $G(x_1 - x_2) = \langle \mathcal{T}\{\Psi^\dagger(x_1)\Psi(x_2)\}\rangle$ [220,221], where the expectation value is calculated over the groundstate with non-zero interactions. Following the prescriptions of quantum-many body physics within condensed matter [77], by going to momentum space, the inverse of the Green's function can be expressed as

$$G^{-1}(ik_0, \boldsymbol{k}) = ik_0 - \mathbb{H}_0(\boldsymbol{k}) + \Sigma_s(ik_0, \boldsymbol{k}), \tag{66}$$

where $\mathbb{H}_0(\boldsymbol{k})$ is the single-particle Hamiltonian, the $N$-flavour version of Eq. (29) in our case, and $\Sigma_s(ik_0, \boldsymbol{k})$ is the so-called self-energy. This self-energy contains all the one-particle irreducible "tadpole" contributions to the fermion propagator arising from intermediate scattering processes in which particle-antiparticle pairs are virtually created from the groundstate. Within

our large-$N$ theory, we have precisely calculated those at leading order in $N$ by introducing the auxiliary fields. In fact, the above condensates can be readily used to approximate this self-energy as

$$\Sigma_{\mathrm{s}}(\mathrm{i}k_0, \boldsymbol{k}) = \mathbb{1}_N \otimes \left( \gamma^0 \gamma^\mu A_\mu + \mathrm{i}\gamma^0 \gamma^3 \Sigma_1 + \mathrm{i}\gamma^0 \gamma^5 \Sigma_2 + \gamma^0 \Pi \right), \tag{67}$$

which has no momentum dependence $\Sigma_{\mathrm{s}}(\mathrm{i}k_0, \boldsymbol{k}) = \Sigma_{\mathrm{s}}(0, \boldsymbol{0})$ since we have assumed the condensates to be homogeneous.

As discussed in [215–219], the static contributions to the self-energy $\Sigma_{\mathrm{s}}(0, \boldsymbol{k})$ can be used to define the so-called topological Hamiltonian. To consider the same symmetry class as the non-interacting one, we set $A = 0$ and $\Pi = 0$, such that the discrete SO(5) rotation is preserved. In this case, the only contribution to the topological Hamiltonian stems from the scalar condensates

$$\mathbb{H}_{\mathrm{t}}(\boldsymbol{k}) = \mathbb{H}_0(\boldsymbol{k}) + \Sigma_{\mathrm{s}}(0, \boldsymbol{k}) = \mathbb{H}_0(\boldsymbol{k}) + \mathrm{i}\gamma^0 \gamma^3 \Sigma_1 + \mathrm{i}\gamma^0 \gamma^5 \Sigma_2. \tag{68}$$

Within this large-$N$ approximation, the many-body topological invariant can be expressed in terms of the two Chern-Simons forms (44) with the twisted Wilson masses renormalised by the two scalar condensates (62), namely

$$\mathsf{W}_j(\Sigma) = \frac{N}{2} \sum_{\ell_j = 0,1} (-1)^{\ell_j} \operatorname{sgn}\{m_j(\boldsymbol{k}_{\mathrm{D}}, \Sigma)\}, \tag{69}$$

The full topological invariant of the correlated HOTI can be approximated, within the large-$N$ limit, as follows

$$e^{\mathrm{i}\pi \mathsf{W}_1(\Sigma)\mathsf{W}_2(\Sigma)} = \begin{cases} (-1)^N, & \text{if } -\frac{2r}{a} < m_j + \Sigma_j < 0, \qquad \forall j, \\ (+1)^N, & \text{else.} \end{cases} \tag{70}$$

Accordingly, the large-$N$ solution obtained by minimizing the effective potential (65) can be readily used to extract $\Sigma_1, \Sigma_2$ in the parameter regime where the discrete SO(5) symmetry is still preserved, and localise the critical surface that separates the correlated HOTI from the trivial band insulator. We recall again that, in order to have the symmetry protection, we need to consider the regime where $m_1 = m_2$ and $\Sigma_1 = \Sigma_2$, which will cut this critical surface determining a critical line.

### 5.4 Large-$N$ results and phase diagram

We start by making the assumption that only the pseudo-scalar auxiliary field condenses $\Pi = \langle \pi \rangle \neq 0$, which we expect describes the leading symmetry-breaking channel among all other condensates, and should set in at a certain value of the interaction strength $g^2 > 0$. This assumption will allow us to derive a set of simple gap equations that can be later used as a reference in the minimization of the full effective potential (65) that contains all possible condensation channels (63). As with gap equations in other models [95, 96, 133], this formalism is applicable whenever the order parameter is non-zero $\Pi > 0$ [214]. As advanced in the previous sections, the $\sigma$ condensates typically attain non-zero values for any point in parameter space $(m_1 a, m_2 a, g^2/a)$, except for the line at fixed twisted mass $m_1 a = m_2 a = r$. We hence consider that $\Sigma = \langle \boldsymbol{\sigma} \rangle \neq 0$. Therefore, the main assumption in the following gap equations is that the vector condensate vanishes $A = \langle \boldsymbol{a} \rangle = 0$ (this will be justified in Fig. 7 below). The gap equations are derived by solving the set of non-linear equations given by the stationary point $\partial_\Pi V_{\mathrm{eff}}|_{A_\mu = 0} = \partial_{\Sigma_1} V_{\mathrm{eff}}|_{A_\mu = 0} = \partial_{\Sigma_2} V_{\mathrm{eff}}|_{A_\mu = 0} = 0$. Setting the Wilson parameter to $r = 1$ henceforth, we find

$$\frac{1}{g^2} = 4 \int \frac{\mathrm{d}k_0}{2\pi} \int_k \frac{1}{k_0^2 + \hat{\boldsymbol{k}}^2 + \boldsymbol{m}^2(\boldsymbol{k}, \Sigma) + \Pi^2}, \tag{71}$$

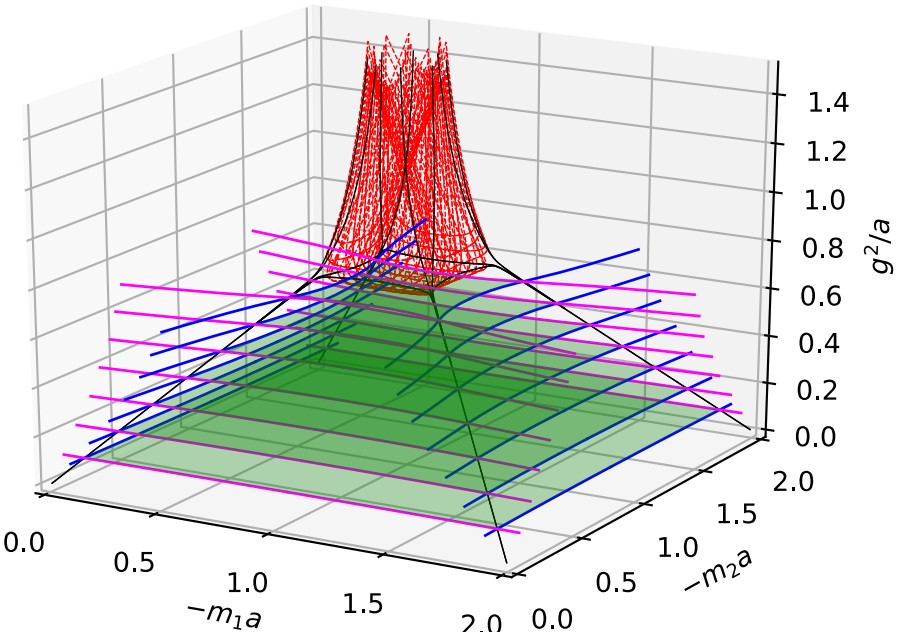

Figure 5: **Interacting higher-order topological phase diagram I:** We consider parameter space $(m_1 a, m_2 a, g^2/a)$, and represent the different phases in the twisted-Wilson lattice model with 4-Fermi interactions. The blue and magenta curves (74) delimit the inner green area (73) at fixed-$g^2$ slices that has a non-trivial many-body topological invariant $\mathrm{e}^{\mathrm{i}\pi W_1(\Sigma)W_2(\Sigma)} = -1$ for an odd number of fermion flavours. The red lines are obtained by solving the gap equations to localise the boundary of the parity-breaking $\Pi$ condensate (71)-(72). The black lines demonstrate that this pseudo-scalar condensate actually grows from the four corners of the non-interacting HOTI phase at $g^2 = 0$ (see Fig. 4), forming the shape of an Eiffel tower that rests on the correlated HOTI. All numerical evaluations employed a spatial lattice with $N_s = 32^2$ sites.

where we have assumed that $\Pi > 0$ to simplify the equation. In addition, we get the following two equations from the derivatives with respect to the scalar condensates

$$\frac{m_j a}{g^2} = -4 \int \frac{\mathrm{d}k_0}{2\pi} \int_k \frac{1 - \cos k_j a}{k_0^2 + \hat{\boldsymbol{k}}^2 + \boldsymbol{m}^2(\boldsymbol{k}, \boldsymbol{\Sigma}) + \Pi^2} \,, \tag{72}$$

where we have used Eq. (71) to simplify them, getting an expression that relates to the twisted masses $m_1, m_2$.

Equations (71)-(72) can be solved with $\Pi = 0^+$ to map out the boundary of the phase hosting a pseudo-scalar condensate. This critical region is a surface in parameter space $(m_1 a, m_2 a, g^2/a)$, and is shown in red in Fig. 5. This surface is spanned by a number of trajectories that represent solutions to the gap equations for $\Pi = 0^+$, for which we fix a different value of the twisted masses renormalised by the scalar condensates $(m_j + \Sigma_j)$. These trajectories are plotted in Fig. 5 as a collection of red dashed curves; the resulting network visualises a closed surface which descends from strong coupling down to $g^2/a \approx 0.7$. There are a couple of interesting comments: *(i)* The volume inside the red surface describes the spontaneous symmetry-broken phase with a pseudo-scalar condensate. This condensate breaks any of the parities, and corresponds to the so-called Aoki phase found in other lattice field theories [213]. As emphasised above, the more important thing is that this pseudo-scalar condensate

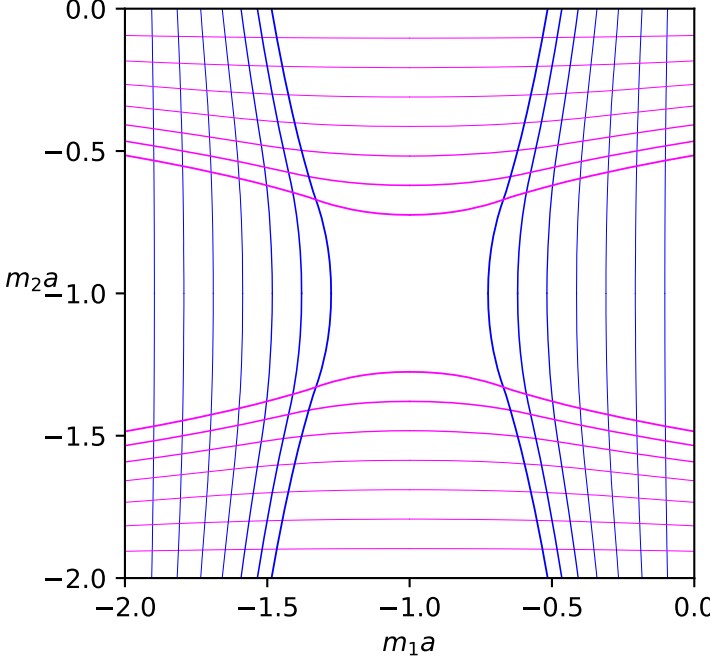

Figure 6: **Interacting higher-order topological phase diagram II:** Here the lines where the gap functions $f_1 = 0$ (blue) and $f_2 = 0$ (magenta) are plotted for increasing values of the coupling strength $g^2/a \in \{0.1, 0.2, 0.3, 0.4, 0.5, 0.6, 0.7\}$, which grow as one moves towards the center of the square. The HOTI phase is contained within a star-shaped region that is delimited by groups of four of these lines, corresponding to the same value of the coupling strength $g^2/a$.

also breaks spontaneously the discrete SO(5) symmetry responsible for the protection of the HOTI. Hence the higher-order topological invariant and the corner modes cannot coexist with this pseudo-scalar condensate. *(ii)* The volume that contains this SO(5) breaking condensate is centered around the symmetry line $m_1 a = m_2 a = -1$ in which the scalar condensates vanish $\boldsymbol{\Sigma} = \mathbf{0}$, and gets more and more compressed as the interaction strength becomes large $g^2/a \gg 1$. On the other hand, for for $g^2 = 0$, the gap equation (71) becomes singular at the four corners $(m_1, m_2) \in \{(0, 0), (0, -2), (-2, 0), (-2, -2)\}$, and we have found that this phase extends all the way down to weak coupling in four very sharp spikes. Let us note that this is a large-$N$ prediction, and a different method should be used to determine the extent of this phase for finite $N$.

As already remarked above, the gap equations (71)-(72) are only valid for $\Pi > 0$. On the other hand, the original question that motivated the study was to see the extent of the HOTI as one increases interactions, which would require exploring the region with $g^2 > 0$ for which $\Pi = 0$, more particularly, the weaker-coupling regime beneath the Aoki phase. To explore it, we will directly minimize the effective potential $V_{\text{eff}}(\mathbf{0}, \boldsymbol{\Sigma}, 0)$ at a specific $(m_1 a, m_2 a, g^2/a)$. As discussed in Eq. (70), the many-body topological invariant for the HOTI in the large-$N$ limit is non-trivial $e^{i\pi W_1(\boldsymbol{\Sigma}) W_2(\boldsymbol{\Sigma})} = -1$ when

$$-2 < m_1 a + \Sigma_1 a < 0, \qquad -2 < m_2 a + \Sigma_2 a < 0, \tag{73}$$

and we have an odd number of flavours. The correlated HOTI phase will then be contained in a region bounded by contours along which a gap function $f_j(k_j a, \Sigma_j a) = m_j a + (1 - \cos k_j a) + \Sigma_j a$ (with $j = 1, 2$) vanishes at either the origin $k_j = 0$, or at the zone edge $k_j = \pi/a$. The

procedure is then to search for solutions of the non-linear equations

$$m_j + \Sigma_j = n_j, \qquad n_j \in \{0, -2\}, \tag{74}$$

using the value of the scalar condensates $\Sigma_1, \Sigma_2$ obtained by numerical minimisation of $V_{\text{eff}}(\mathbf{0}, \Sigma, 0)$. In practice, we define a circle centered at the line of symmetry where $\Sigma = \mathbf{0}$ by defining the twisted masses as $(m_1, m_2) = (-1/a, -1/a) + m(\cos\theta, \sin\theta)$, for $m > 0$ and $\theta \in [0, 2\pi)$. We then search for the roots of Eq. (74) by scanning first in $\theta$ and, subsequently, in $m$. The resulting contours along which a gap function vanishes $f_j(k_j, \Sigma_j) = 0$ are plotted as blue (magenta) lines in Figs. 5,6 for various values of the coupling strength $g^2$. Each pair of blue (magenta) lines for a fixed coupling strength corresponds to the renormalised twisted mass proportional to $i\gamma^3$ ($i\gamma^5$) satisfying $f_1(0, \Sigma_1) = 0$ or $f_1(\pi/a, \Sigma_1) = 0$ ($f_2(0, \Sigma_2) = 0$ or $f_2(\pi/a, \Sigma_2) = 0$). The region of the correlated HOTI phase is enclosed within the areas inside the four intersecting lines, and is depicted in Fig. 5 by a shaded green area that connects to the green square in the non-interacting limit (Cf. Fig. 4), and projected onto the $(m_1, m_2)$-plane in Fig. 6. As $g^2$ increases, the borders of this region curve inwards, and the HOTI shrinks until it roughly coincides with the lateral extent of the Aoki phase at the critical coupling $g^2/a \simeq 0.7$ (note the lower surface of the Aoki phase has convex curvature, confirmed in Fig. 7). On the other hand, all the empty region surrounding both the green and red volumes corresponds to a trivial band insulator, in which the topological invariant is trivial and the pseudo-scalar condensate vanishes. All the critical surfaces predicted within our large-$N$ methods correspond to higher-order quantum phase transitions, either topological or symmetry-breaking ones.

Once we have discussed the phase diagram of Fig. 5 in detail, we should check for the consistency of the assumptions we made about the competing condensates by minimising the full effective potential $V_{\text{eff}}(A, \Sigma, \Pi)$. We recall again that one can set $A_0 = 0$ as discussed in Appendix D, but must consider the other competing condensation channels along lines of fixed $(m_1, m_2)$. Fig. 7 shows the resulting condensates for two choices of $(m_1, m_2)$ at different distances from the line of symmetry $m_1 = m_2 = -1/a$. The $\Pi$ condensate signalling the SO(5)-breaking Aoki phase rises from zero at a critical coupling $g_c^2/a \approx 0.7$, and the opposite signs of the scalar condensates $\Sigma_1 = -\Sigma_2$ correspond to $(m_1, m_2)$ lying on opposite sides of the symmetric line $(-1/a, -1/a)$. Crucially, the current condensate $A_1$ remains zero throughout, justifying our previous assumption where we set $A = \mathbf{0}$. Solutions with $A \neq \mathbf{0}$, as found for an interacting Chern-insulator two-band model in [95, 96], could only be found in our lattice model by artificially constraining $\Pi = 0$. Otherwise, we find that the SO(5)-breaking Aoki condensate always describes the leading condensation channel $\Pi > 0$ as one increases the coupling strength. Further from the line of symmetry, as shown by the dashed lines in Fig. 7 the picture remains qualitatively the same, but with values of the scalar condensates $\Sigma_i$ that become larger in magnitude; this time at strong-enough coupling the trajectory actually re-enters the SO(5)-symmetric phase, reflected by the renewed vanishing of $\Pi$ and the kinks in $\Sigma_i(g^2)$. however, rather than re-entering into the HOTI phase, one goes into a trivial band insulator where, even if the SO(5) symmetry is preserved, the topological invariant is trivial and the zero states are no longer localised at the corners of the system.

## 6 Conclusion and outlook

In this work, we have presented a non-standard lattice regularization of Dirac QFTs based on a new type of Wilson fermions that have an anisotropic twisted Wilson mass. We have shown that the anisotropic twisted Wilson mass is responsible for the occurrence of HOTIs that display zero-energy corner modes, and a non-vanishing topological invariant in the bulk. We have discuss a cold-atom implementation of this lattice field theory that exploits Raman optical lattices, and spin-3/2 Fermi gases of alkaline-earth atoms. Interestingly, the *s*-wave scattering

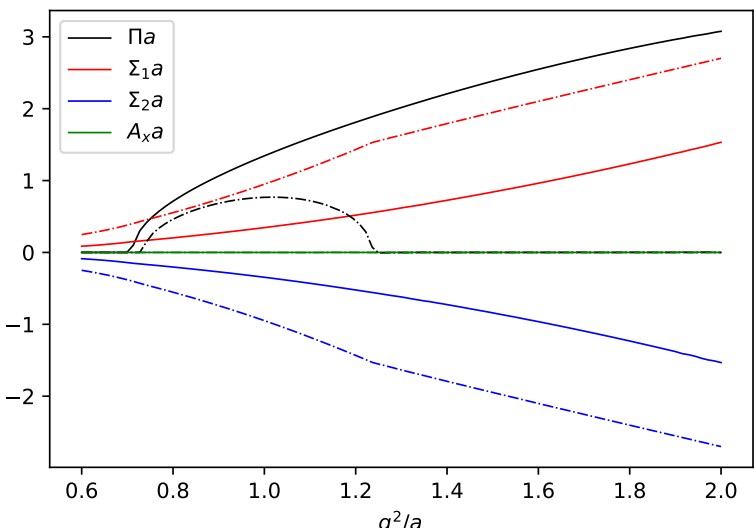

Figure 7: **Competing condensation channels:** Condensates as a function of $g^2$ for $(m_1 a, m_2 a) = (-0.95, -1.05)$ (full lines) and $(-0.85, -1.15)$ (dot-dashed lines). The opposite signs of $\Sigma_1, \Sigma_2$ mirror the signs of $-1 - m_{1,2}a$. Note that $A = 0$ throughout.

of this atoms leads to a SO(5) invariant 4-Fermi interaction, which leads to an interesting competition of HOTI phases and various fermion condensates. We explored the full phase diagram of the model using large-$N$ techniques, and argued that the correlated HOTI phase eventually gives way to a parity-breaking fermion condensate.

Since the microscopic parameters of the model can be independently tuned in the proposed cold-atom experiment, it would be very interesting that the predictions presented in this work could be tested in future experiments. We note that the recent experimental work on the quantum simulator of Chern insulators using $^{87}$Sr Fermi gases in Raman optical lattices [159] is very promising in this direction. If one can use these methods for a different alkaline-earth atoms such as $^{132}$Cs, controlling the four spin states as discussed in our work, the experiment would directly probe the SO(5) self-interacting Dirac QFT that has been the subject of our work. This experiment, together with related theory works [222, 223], have also shown that it is possible to perform certain measurements to infer the value of a topological invariant similar to the Chern number. It would be interesting to explore if such methods can also be adapted to the spin-3/2 Fermi gas, and used to infer the value of our higher-order topological invariant. Other than that, further studies are required in order to propose other measurement schemes. For instance, in order to retrieve the order parameter associated to the fermion condensates discussed in this work, some sort of spin-resolved imaging by either illuminating the gas and processing its shadow, or using quantum gas microscopes, should be required. These methods should be combined with microwave transition and spin-selective techniques to infer the atomic densities corresponding to the order parameters.

From a more theoretical perspective, it would be very interesting to explore finite-temperature and finite-density phases in this model of SO(5) interacting Dirac matter. In particular, by moving away from half filling, one should consider other possible condensation channels that likely include superconducting and inhomogeneous orders, or even new exotic orders that go beyond the Landau symmetry breaking paradigm. The study of those drawing further connections between high-energy physics, condensed matter, and AMO physics, will contribute to the growing interest in this interdisciplinary line of research [224–232].

# Acknowledgments

**Funding information** A. B. acknowledges support from PID2021-127726NB-I00 (MCIU/AEI/FEDER, UE), from the Grant IFT Centro de Excelencia Severo Ochoa CEX2020-001007-S, funded by MCIN/AEI/10.13039/501100011033, and from the CSIC Research Platform on Quantum Technologies PTI-001. S.H. was supported by the STFC Consolidated Grant ST/T000813/1. D.G.-C. is supported by the Simons Collaboration on Ultra-Quantum Matter, which is a grant from the Simons Foundation (651440, P.Z.).

# A  Clifford algebra and SO($D$) Dirac fermions

In this appendix, we review several aspects that appear in the description of relativistic QFTs of Dirac fermions in arbitrary spatial dimensions and Euclidean imaginary time. We discuss the role of the special orthogonal group of transformations in the description of both spacetime and internal symmetries. We start from the partition function of a Dirac field of mass $m$ which, in natural units $\hbar = c = 1$, can be written as a functional integral $Z = \int [\mathrm{D}\overline{\psi}\,\mathrm{D}\psi]\mathrm{e}^{-S_0}$, provided that the fermionic field $\psi(x)$ and the adjoint $\overline{\psi}(x)$ are described by mutually anti-commuting Grassmann fields with anti-periodic boundary conditions in the imaginary-time direction [77, 78]. The partition function thus depends on the Euclidean action

$$S_0 = \int \mathrm{d}^D x \overline{\psi}(x)\big(\gamma^\mu \partial_\mu + m\big)\psi(x), \tag{A.1}$$

where we have introduced $\mu \in \{0, \cdots, D-1\}$ to label the Euclidean spacetime coordinates $x = (\tau, \boldsymbol{x})$ and derivatives $\partial_\mu = \partial/\partial x^\mu$, using Einstein's criterion of repeated index summation. We have also introduced the gamma matrices $\{\gamma^\mu\}$, which are the generators of the Clifford algebra with Euclidean metric $\mathrm{Cl}(0, D)$ [79, 80], and must thus fulfill

$$\{\gamma^\mu, \gamma^\nu\} = \gamma^\mu \gamma^\nu + \gamma^\nu \gamma^\mu = 2g^{\mu\nu}\mathbb{1}_{d_s}, \tag{A.2}$$

where the Euclidean metric $g^{\mu\nu} = \delta^{\mu,\nu} = \delta^\mu_\nu = \delta_{\mu,\nu}$ is defined through the Kronecker delta, and $\mathbb{1}_{d_s}$ is the identity matrix. The Euclidean gamma matrices are thus mutually anti-commuting, all square to the identity, and do not have any distinction between upper and lower indexes $\gamma^\mu = \gamma_\mu$.

In an irreducible representation, these generators can be expressed by square Hermitian matrices $\gamma_\mu \in \mathrm{Herm}(\mathbb{C}^{d_s})$ acting on a vector space of dimension $d_s = 2^{\lfloor D/2 \rfloor}$, where $\lfloor x \rfloor$ stands for the greatest integer less than or equal to $x$. Accordingly, for spacetime dimensions $D = 2n$ or $D = 2n + 1$ with $n \in \mathbb{Z}^+$, they are $2^n \times 2^n$ Hermitian matrices that can be built from specific tensor products within the orthogonal Pauli basis $\gamma^\mu \in \mathcal{B}_n = \{\mathbb{1}_2, \sigma^x, \sigma^y, \sigma^z\}^{\otimes n}$. As discussed for instance in reference [79], there are simple recipes to construct these gamma matrices, and the only difference between odd $D = 2n + 1$, and even $D = 2n$ spacetime dimensions is that the latter has an additional $\gamma^{2n} = \gamma_\star$, obtained by multiplying all the matrices of $D = 2n$ together $\gamma_\star = (-\mathrm{i})^n \gamma^0 \gamma^1 \cdots \gamma^{2n-1}$.

Once the gamma matrices are known, we can obtain the $2^D$ elements of the Clifford algebra using products. In particular, the anti-symmetric products leads to elements of the Clifford algebra being ordered according their rank as

$$\begin{aligned}
\mathrm{Cl}(0, 2n) &= \big\{\mathbb{1}_{d_s}, \gamma^{\mu_1}, \gamma^{\mu_1\mu_2}, \gamma^{\mu_1\mu_2\mu_3}, \cdots, \gamma^{\mu_1\mu_2\cdots\mu_{2n}}\big\}_{\mu_j=0}^{2n-1}, \\
\mathrm{Cl}(0, 2n+1) &= \big\{\mathbb{1}_{d_s}, \gamma^{\mu_1}, \gamma^{\mu_1\mu_2}, \gamma^{\mu_1\mu_2\mu_3}, \cdots, \gamma^{\mu_1\mu_2\cdots\mu_{2n}}\big\}_{\mu_j=0}^{2n},
\end{aligned} \tag{A.3}$$

where $\gamma^{\mu_1\mu_2} = \frac{1}{2}[\gamma^{\mu_1}, \gamma^{\mu_2}]$, and the remaining are defined recursively $\gamma^{\mu_1\mu_2\cdots\mu_j} = \frac{1}{2}[\gamma^{\mu_1}, \gamma^{\mu_2\cdots\mu_j}]$, for $j \in \{3, \cdots, D\}$.

In the context of the Dirac QFT, the elements of the Clifford group with rank 1, 2 play a key role. The rank-1 elements, namely the aforementioned gamma matrices, enter in the definition of the action (A.1). The rank-2 elements $\{\gamma^{\mu_1\mu_2}\}$, which correspond to the $D(D-1)/2$ anti-symmetric products of the gamma matrices, serve as the generators of the spacetime rotations $x^\mu \mapsto \Lambda^\mu_\nu x^\nu$ with $\Lambda \in SO(D)$, which would correspond to Lorentz transformations $SO(1, D-1)$ if we rotated back to real time $\tau \to it$ [1]. Indeed, any such transformation $\Lambda$ has a representation in terms of a rotation of angle $\theta_{\mu\nu}$ within the $(\mu\nu)$-plane and the infinitesimal generator $\gamma^{\mu\nu}$. This so-called spinor representation reads $S_\Lambda = \exp\{\frac{1}{4}\omega_{ab}\gamma^{ab}\}$ where $\omega_{ab} = \theta_{ab}(\delta^a_\mu \delta^b_\nu - \delta^a_\nu \delta^b_\mu)$, which can be easily checked to yield a unitary representation of the $SO(D)$ group $S_\Lambda^{-1} = (S_\Lambda)^\dagger$. This leads to a crucial difference with respect to Minkowski spacetime, and forbids defining the adjoint as $\overline{\psi}(x) = \psi^\dagger(x)\gamma_0$. The field and its adjoint are mutually anti-commuting Grassmann spinors with an even number of components $d_s$, and one postulates that they transform under spacetime rotations as

$$\psi(x) \mapsto S_\Lambda \psi\left(\Lambda^{-1}x\right), \overline{\psi}(x) \mapsto \overline{\psi}\left(\Lambda^{-1}x\right)S_\Lambda^{-1}. \tag{A.4}$$

One then finds that the Euclidean action (A.1) is invariant under $SO(D)$, which also requires using the transformations of the rank $j < D$ Clifford elements as tensors under $SO(D)$

$$S_\Lambda^{-1}\gamma^{\mu_1\mu_2\cdots\mu_j}S_\Lambda = \Lambda^{\mu_1}_{\nu_1}\Lambda^{\mu_2}_{\nu_2}\cdots\Lambda^{\mu_j}_{\nu_j}\gamma^{\nu_1\nu_2\cdots\nu_j}. \tag{A.5}$$

This shows that the gamma matrices transform as a vector under $SO(D)$, such that $\overline{\psi}\psi$ and $\overline{\psi}\gamma^a\partial_a\psi$ are scalars, and the Euclidean action is invariant under $SO(D)$.

For even spacetime dimensions $D = 2n$, the highest-rank element of the Clifford algebra (A.3) also plays an important role. It can be used to define an additional Hermitian matrix that anti-commutes with all the spacetime gamma matrices, and is thus left invariant under any $SO(2n)$ rotation

$$\gamma_\star = (-i)^n\gamma^0\gamma^1\cdots\gamma^{2n-1} \mapsto S^{-1}(\Lambda)\gamma_\star S(\Lambda) = \gamma_\star. \tag{A.6}$$

In $D = 4$, this is typically called the chiral gamma matrix $\gamma^5 = -\gamma^0\gamma^1\gamma^2\gamma^3$, which can be used to decompose the Dirac spinor into left- and right-handed two-component spinors, the so-called chiral Weyl fermions [1]. Alternatively, in any even spacetime dimension, this gamma matrix can serve to propose a twisting of the scalar mass

$$S_0 = \int d^D x \overline{\psi}(x)\left(\gamma^\mu\partial_\mu + m_1 + im_2\gamma_\star\right)\psi(x), \tag{A.7}$$

where $m_1 = m\cos\theta$, and $m_2 = m\sin\theta$. The anti-commuting mass terms can be expressed as

$$me^{i\theta\gamma_\star}\overline{\psi}\psi = m\cos\theta\overline{\psi}\psi + im\sin\theta\overline{\psi}\gamma_\star\psi, \tag{A.8}$$

which respects Lorentz $SO(2n)$ invariance according to Eqs. (A.4) and (A.6), and can be seen as the result of an axial rotation $\psi \mapsto \exp\{i\frac{\theta}{2}\gamma_\star\}\psi$, $\overline{\psi} \mapsto \overline{\psi}\exp\{i\frac{\theta}{2}\gamma_\star\}$. On the other hand, the second one breaks explicitly the parity symmetry since

$$\left.\begin{array}{l}\mathcal{P}\psi(\tau, \boldsymbol{x}) = \gamma^0\psi(\tau, -\boldsymbol{x}) \\ \mathcal{P}\overline{\psi}(\tau, \boldsymbol{x}) = \overline{\psi}(\tau, -\boldsymbol{x})\gamma^0\end{array}\right\} \implies \left.\begin{array}{l}\overline{\psi}\psi \mapsto \overline{\psi}\psi \\ \overline{\psi}\gamma_\star\psi \mapsto -\overline{\psi}\gamma_\star\psi\end{array}\right\}. \tag{A.9}$$

For odd spacetime dimensions $D = 2n + 1$, this $\gamma_\star$ matrix plays the role of the gamma matrix for the new spatial direction $\gamma^{2n} = \gamma_\star$. Therefore, the product of all spacetime gamma

matrices is trivial $\gamma^0\gamma^1\cdots\gamma^{2n} \propto \mathbb{1}_{d_s}$, and the SO($2n+1$)-invariant Dirac action for free Dirac fields can only take the form of Eq. (A.1). Therefore, only the standard mass term $m\overline{\psi}\psi$ can be considered which, as discussed in the following section would break the invariance under the corresponding parity transformation. In the following subsection, we will explain how to go beyond these limitations when considering a reducible representation of the Clifford algebra.

## B  Dimensional reduction and 4-Fermi interactions

Motivated by the experimental situations discussed in the main text, we can also consider reducible representations of the Clifford algebra for a specific spacetime dimension. In this appendix, we will consider odd dimension $D = 2n + 1$, and understand a reducible representation of the Clifford algebra as a consequence of an effective dimensional reduction. We consider $2n + 3$ dimensions initially, where the spinor dimension is doubled with respect to the irreducible one of $D = 2n + 1$, and we get two additional gamma matrices $\gamma^{2n+1}$, and $\gamma_\star = \gamma^{2n+2}$. We will label the higher-dimensional spacetime coordinates with latin indexes $a \in \{0, 1, \cdots, 2n + 2\}$. The SO($2n + 3$)-invariant action is that of Eq. (A.1), and we will focus in the massless case $m = 0$, namely

$$S_0 = \int \mathrm{d}^D x\, \overline{\psi}(x)\gamma^a \partial_a \psi(x). \tag{B.1}$$

We can rewrite this action by separating the contribution of the two extra spatial dimensions

$$S_0 = \int \mathrm{d}^{2n+3} x\, \overline{\psi}(x)\big(\gamma^\mu \partial_\mu + \gamma^{2n+1}\partial_{2n+1} + \gamma_\star \partial_{2n+2}\big)\psi(x), \tag{B.2}$$

where the index $\mu \in \{0, \cdots, 2n\}$ is restricted to the lower number of dimensions $D = 2n + 1$ in the reduced spacetime.

As discussed in [82], the dimensional reduction is inspired by the Kaluza-Klein compactification [81], and proceeds in two steps. In the first one, the $x_{2n+2}$ spatial direction is compactified to a circle $x_{2n+2} + r = x_{2n+2}$ with a very small radius $r \to 0$. Considering that the Grassmann fields are periodic in the spatial direction, the corresponding momentum $p_{2n+2} = -\mathrm{i}\partial_{2n+2} \propto \ell_{2n+2}/r$ gets quantised in terms of the integers $\ell_{2n+2} \in \mathbb{Z}_{N_{2n+2}}$, one readily sees that only the quantum number $\ell_{2n+2} = 0$ plays a role in the low-energy physics as $r \to 0$. From the perspective of the non-compact dimensions, one gets a tower of very heavy Dirac fields, and focusing on low energies amounts to a truncation of such high-energy modes [80]. In the presence of an additional scalar field $\sigma_\star(x)$ that is minimally coupled to the fermions $\partial_{2n+2} \to \partial_{2n+2} + \mathrm{i}\sigma_\star(x)$, and assuming that this scalar field is homogeneous $\sigma_\star(x) = m_\star$, the dimensional reduction leads to an effective low-energy action action that reads

$$S_0 = \int \mathrm{d}^{2n+2} x\, \overline{\psi}(x)\big(\gamma^\mu \partial_\mu + \gamma^{2n+1}\partial_{2n+1} + \mathrm{i}m_\star \gamma_\star\big)\psi(x), \tag{B.3}$$

which is now invariant under Lorentz transformations in the reduced spacetime, such that SO($2n + 3$) $\mapsto$ SO($2n + 2$).

In a second step, we compactify the $x_{2n+1}$ direction, introducing also a minimally-coupled scalar field $\sigma_{2n+1}(x)$. Projecting again onto the low-energy physics when $r \to 0$, and thus considering only the $\ell_{2n+1} = 0$ quantised momentum for a homogeneous field $\sigma_{2n+1}(x) = m_{2n+1}$, we are led to

$$S_0 = \int \mathrm{d}^D x\, \overline{\psi}(x)\big(\gamma^\mu \partial_\mu + \mathrm{i}m_{2n+1}\gamma^{2n+1} + \mathrm{i}m_\star \gamma_\star\big)\psi(x), \tag{B.4}$$

where $SO(2n+2) \mapsto SO(2n+1)$. By comparing this dimensionally-reduced action to that of Eq. (A.1), we see that the sigma fields play a similar role to the mass terms when they are homogeneous. The main difference is that we have more freedom in the definition of parity, and these two mass terms can open a gap in the parity-symmetric case. For odd spacetime dimensions, parity must be understood as a transformation that reverses only an odd number of the spatial directions, for example

$$
\begin{aligned}
\mathcal{P}\psi(\tau,\boldsymbol{x}) &= \gamma^{2n}\psi(\tau,x_1,\cdots x_{2n-1},-x_{2n}), \\
\mathcal{P}\overline{\psi}(\tau,\boldsymbol{x}) &= -\overline{\psi}(\tau,x_1,\cdots x_{2n-1},-x_{2n})\gamma^{2n},
\end{aligned}
\tag{B.5}
$$

such that both mass terms (B.4) are invariant under parity

$$
\begin{aligned}
\overline{\psi}\gamma^{2n+1}\psi &\mapsto \overline{\psi}\gamma^{2n+1}\psi, \\
\overline{\psi}\gamma_\star\psi &\mapsto \overline{\psi}\gamma_\star\psi.
\end{aligned}
\tag{B.6}
$$

Such a transformation can be defined for any other spatial axis, or a combination of an odd number of them.

We note again that this effective action (B.4) is $SO(2n+1)$ invariant in the reduced spacetime, but there is a higher $SO(2n+3)$ invariance if one considers rotations in the full spacetime with the two additional compactified dimensions. The important point is that, as a remnant of the compactified dimensions, the spinors inherit the dimensionality given by the corresponding representation of the higher-dimensional Clifford algebra (see Appendix A). This enlarged number of spinor components and bigger symmetry group can play an important role once we introduce interactions. Indeed, we can also consider introducing $SO(2n+3)$ invariant quartic interactions, which can be obtain by considering the transformations of fermionic bilinears built from the Clifford algebra elements (A.5). In particular, we can add an $SO(2n+3)$-invariant 4-Fermi term to the action

$$
S_{\text{int}} = \int \mathrm{d}^D x \frac{g^2}{2}\Big(-(\overline{\psi}\psi)^2 - (\overline{\psi}\mathrm{i}\gamma^a\psi)(\overline{\psi}\mathrm{i}\gamma_a\psi)\Big).
\tag{B.7}
$$

The first term corresponds to the so-called Gross-Neveu interaction [7], and is a scalar under the $SO(2n+3)$ Lorentz transformations. The other $2n+3$ terms are quartic interactions corresponding to the so-called Thirring term [83], which is a vector-vector interaction that is also invariant under $SO(2n+3)$ Lorentz transformations. On the other hand, if one rewrites these terms as $(\overline{\psi}\mathrm{i}\gamma^a\psi)^2 = (\overline{\psi}\mathrm{i}\gamma^\mu\psi)^2 - (\overline{\psi}\gamma^{2n+1}\psi)^2 - (\overline{\psi}\gamma_\star\psi)^2$, and reinterprets them from the perspective of the dimensionally-reduced spacetime, only the $(\overline{\psi}\gamma^\mu\psi)^2$ terms correspond to the squared magnitude of a vector under $SO(2n+1)$, whereas the two additional terms $(\overline{\psi}\mathrm{i}\gamma^{2n+1}\psi)^2$ and $(\overline{\psi}\mathrm{i}\gamma_\star\psi)^2$ are scalars. We emphasise that these additional terms are only allowed by the fact that we are working with a reducible representation of the gamma matrices, which are allowed by the larger number of spinor degrees of freedom in the enlarged spacetime. In the main text, we have referred to the action $S_0 + S_{\text{int}}$ in Eqs. (B.3) and (B.7) for $n=1$ as our model of Dirac matter with $SO(2n+3)=SO(5)$ 4-Fermi interactions. We discuss possible lattice regularizations that allow us to discuss higher-order topological phases and competing symmetry-breaking condensates as one increases the strength of the quartic interactions. This regularization requires rewriting the dimensionally-reduced masses in Eq. (B.4) in terms of twisted Wilson masses, as discussed in the main text.

## C Standard Wilson fermions and Chern insulators

In this Appendix, we present the details for a standard Wilson discretization of the reducible Dirac QFT in Eq. (1). As noted in the main text, this regularization amounts to the introduction

of a momentum-dependent shift $m_1 \mapsto \overline{m}_1(\boldsymbol{k})$ of one of the masses in Eq. (5), whereas the other mass is set to zero $m_2 \mapsto \overline{m}_2 = 0$. The Wilson mass depends on a real parameter $r$ as follows

$$\overline{m}_1(\boldsymbol{k}) = m_1 + \frac{r}{a}\Big(2 - \cos(k_1 a) - \cos(k_2 a)\Big), \tag{C.1}$$

which can be understood as the consequence of a finite-difference discretizations of terms involving higher-order spatial derivatives [86]. We note that we have used an overline in the function in order to differentiate this standard Wilson mass from the twisted Wilson mass in Eq. (9)

For this regularised model in $D = 2 + 1$ dimensions, one actually finds that it corresponds to two copies of the square-lattice version [93–96] of Haldane's quantum anomalous Hall effect [92], leading to a Chern insulator. The full band structure consists of four energy bands with a two-fold degeneracy $\epsilon_{q,\pm}(\boldsymbol{k}) = \pm\epsilon(\boldsymbol{k})$ for $q \in \{1, 2\}$, where

$$\epsilon(\boldsymbol{k}) = \sqrt{\hat{\boldsymbol{k}}^2 + \overline{m}_1^2(\boldsymbol{k})}. \tag{C.2}$$

The groundstate is then obtained by filling all the negative energy states $|\mathrm{gs}\rangle = \prod_{k \in \mathrm{BZ}} \prod_{q=1,2} |\epsilon_{q,-}(\boldsymbol{k})\rangle$, and one clearly sees from the above dispersion that the Wilson term leads to a different mass for each Dirac point (7), namely

$$\overline{m}_1(\boldsymbol{k}_{\mathrm{D},\ell}) = m_1 + \frac{r}{a}\Big(2 - (-1)^{\ell_1} - (-1)^{\ell_2}\Big). \tag{C.3}$$

For notational convenience, we can define a mass matrix that contains the four Wilson masses

$$\overline{M}_{\mathrm{W},1} = \sum_{\ell \in \mathbb{Z}_2 \times \mathbb{Z}_2} \overline{m}_1(\boldsymbol{k}_{\mathrm{D},\ell})|\ell\rangle\langle\ell| : \quad \langle\ell|\ell'\rangle = \delta_{\ell,\ell'}. \tag{C.4}$$

As outlined above, by setting $m_1 = 0$, one sees that the spurious doublers become very heavy with a mass of the order of the lattice cutoff $\overline{m}_1(\boldsymbol{k}_{\mathrm{D},\ell}) \propto r/a, \forall \ell \neq \boldsymbol{0}$. On the contrary, the fermion at the origin of the BZ remains massless; $\overline{m}_1(\boldsymbol{k}_{\mathrm{D},\boldsymbol{0}}) = 0$. Making a long-wavelength expansion around this point $\boldsymbol{k} \mapsto \boldsymbol{k}_{\mathrm{D},\boldsymbol{0}} + \boldsymbol{k}$ for $|\boldsymbol{k}| \ll \Lambda_{\mathrm{c}}$ now yields a long-wavelength action that coincides with Eq. (1) for $\overline{m}_2 = 0$. We remark that, although the lattice discretization breaks explicitly the invariance under SO(3) Lorentz transformations, one recovers it in the continuum limit around $\boldsymbol{k}_{\mathrm{D},\boldsymbol{0}}$.

Let us finally discuss the connection of the groundstate of this lattice QFT to standard first-order topological insulators, in particular, to the so-called Chern insulators. For the explicit choice of the gamma matrices (2)-(3), there is a block structure that can be exploited to find a basis in which the problem reduces to a pair of decoupled Chern insulators. Indeed, one can prove that the groundstate corresponding to the above Dirac sea can be characterised by a non-vanishing Chern number for the principal $U(1)$ bundle associated to the filled bands [233]. This topological invariant can be expressed as the integral of the Berry curvature $\mathcal{F}_{\mathrm{b},q}^{ij}(\boldsymbol{k}) = \partial_{k_j}\mathcal{A}_q^i(\boldsymbol{k}) - \partial_{k_i}\mathcal{A}_q^j(\boldsymbol{k})$, where the connection is $\mathcal{A}_q^i(\boldsymbol{k}) = -\mathrm{i}\langle\epsilon_{q,-}(\boldsymbol{k})|\partial_{k_i}|\epsilon_{q,-}(\boldsymbol{k})\rangle$ [234]. One can then show that, for our Wilson-fermion QFT, the Chern number is

$$\mathrm{Ch} = \frac{1}{4\pi}\sum_q \int \mathrm{d}k_i \wedge \mathrm{d}k_j \mathcal{F}_{\mathrm{b},q}^{ij}(\boldsymbol{k}) = \frac{1}{\pi}\sum_q \arg\left\{\mathrm{Det}\big(\overline{M}_{\mathrm{W},1}\big)\right\}, \tag{C.5}$$

which thus attains a non-zero value when we have an odd number of Dirac points that carry a negative Wilson mass

$$\mathrm{Ch} = \sum_\ell (-1)^{\ell_1 + \ell_2}\mathrm{sgn}\big(\tilde{m}_1(\boldsymbol{k}_{\mathrm{D},\ell})\big). \tag{C.6}$$

In light of Eq. (C.3), we thus see that, whenever $m_1 a \in (-4r, -2r) \cup (-2r, 0)$, there is a non-vanishing Chern number $\mathsf{Ch} = \pm 2$, signalling that we have two copies of the standard Chern insulator $\mathsf{Ch} = \pm 1$, each corresponding to the square-lattice version [93–96] of Haldane's quantum anomalous Hall effect [92]. Even if there is no net external magnetic field piercing the spatial lattice, the system displays a quantised Hall conductance that is related to the Chern number as in the standard quantum Hall effect [235]. The bulk-boundary correspondence links these topological invariants to the appearance of circulating edge states localised at the spatial boundaries, which are in fact low-dimensional versions of Kaplan's domain-wall fermions in lattice field theories [112–114].

# D  Calculation of the effective SO(5) potential

Remarkably, all resummations required for the calculation of the effective potential $V_{\mathrm{eff}}(\boldsymbol{\Phi})$ are already encountered in the "vanilla" Gross-Neveu model in $2+1$ dimensions. We can rewrite this QFT in terms of an Euclidean Lagrangian containing just a single bosonic auxiliary field

$$\mathcal{L} = \overline{\psi}\big(\mathrm{i}\gamma^\mu p_\mu + m + \phi\big)\psi + \frac{N}{2g^2}\phi^2\,, \tag{D.1}$$

where $\phi(x)$ will be latter identified with the various components of the completing condensates $\boldsymbol{\Phi}$ in our problem. In the large-$N$ limit for a condensate $\boldsymbol{\Phi} = \langle\phi\rangle$, $V_{\mathrm{eff}}$ then contains the sum of all diagrams with a single fermion loop and $n$ external $\phi$ legs [204] (see Fig. 8) which contribute with

$$\begin{aligned}
\frac{V_{\mathrm{eff}}(\boldsymbol{\Phi})}{N} &= \frac{\Sigma^2}{2g^2} + \sum_{n=1}^{\infty}\frac{1}{n}\int_p \mathrm{tr}\left(\frac{-\boldsymbol{\Phi}}{\mathrm{i}p^\mu\gamma_\mu + m}\right)^n \\
&= \frac{\boldsymbol{\Phi}^2}{2g^2} + \sum_n \frac{1}{n}\int_p\left(\frac{-\boldsymbol{\Phi}}{p^2 + m^2}\right)^n \mathrm{tr}I_n\,,
\end{aligned} \tag{D.2}$$

with $I_n = (-\mathrm{i}p^\mu\gamma_\mu + m)^n$ obeying the recurrence

$$I_n = 2mI_{n-1} - (p^2 + m^2)I_{n-2}\,. \tag{D.3}$$

We now use $\mathrm{tr}I_0 = 4$ and $\mathrm{tr}I_1 = 4M$ to deduce the general result

$$\mathrm{tr}I_n = 4\sum_{k=0}^{n}A_{nk}m^k(p^2 + m^2)^{\frac{n-k}{2}}\,, \tag{D.4}$$

where for $n$ even, $k$ is an even integer and the introduced matrix elements are

$$A_{n0} = (-1)^{\frac{n}{2}}\,, \tag{D.5}$$

$$A_{nk} = \mathcal{A}_k(-1)^{\frac{n}{2}}(\tfrac{n}{2})^2\big((\tfrac{n}{2})^2 - 1^2\big)\dots\big((\tfrac{n}{2})^2 - (\tfrac{k}{2} - 1)^2\big)\,,$$

while for $n$ odd, $k$ is odd and

$$A_{nk} = \frac{1}{2}\big(A_{n+1,k+1} + A_{n-1,k+1}\big)\,. \tag{D.6}$$

The constant $\mathcal{A}_k$ in these expressions is defined such a way that one recovers $A_{nn} = 2^{n-1}$.

Calculation of the sum in (D.2) proceeds by considering a resummation of three different cases.

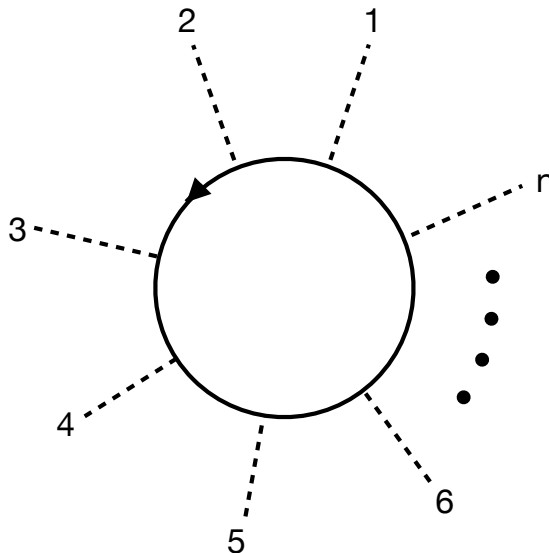

Figure 8: **Large-$N$ effective potential:** Example of a Feynman diagram with $n$ external zero-momentum auxiliary legs (dashed lines), and a single fermion loop (solid circle), yielding the main contribution to the effective potential $V_{\text{eff}}$ in the large-$N$ limit.

*(a) n even, and $k = 0$:* In this case, we find

$$\sum_{n=2,4,\ldots} \frac{4(-1)^{\frac{n}{2}}}{n} \int_p \left( \frac{-\Phi}{\sqrt{p^2+m^2}} \right)^n = \sum_q \frac{2}{q} \int_p \left( \frac{-\Phi^2}{p^2+q^2} \right)^q = -2 \int_p \log\left( 1 + \frac{\Phi^2}{p^2+m^2} \right), \quad \text{(D.7)}$$

where we have made use of the index $q = n/2$.

*(b) n even and k even:* When the integer $k$ is allowed to be even, we reindex using $n = 2q$ and $k = 2\ell$ to find

$$\sum_{q\ell} \frac{2A_{2q,2\ell}}{m} \int_p \left( \frac{\Phi^2}{p^2+m^2} \right)^q \left( \frac{m^2}{p^2+m^2} \right)^\ell. \quad \text{(D.8)}$$

We now use the identity

$$\sum_q A_{2q,2\ell} \frac{z^q}{q} = \frac{2^{2\ell-1}}{\ell} \frac{z^\ell}{(1+z)^{2\ell}}, \quad \text{(D.9)}$$

to perform the following resummation

$$\sum_\ell \frac{1}{\ell} \int_p \frac{(2\Phi m)^{2\ell}}{(p^2+m^2+\Phi^2)^{2\ell}} = -\int_p \log\left( 1 - \frac{4\Phi^2 m^2}{(p^2+m^2+\Phi^2)^2} \right). \quad \text{(D.10)}$$

*(c) n odd and k odd:* When both integers are odd, we use Eq. (D.6) to write the contribution as

$$\sum_{n,k=1,3,\ldots} \frac{4}{n} \int_p \left( \frac{-\Phi}{\sqrt{p^2+m^2}} \right)^n \left( \frac{m}{\sqrt{p^2+m^2}} \right)^k A_{nk}, \quad \text{(D.11)}$$

where the matrix elements fulfill the following identity

$$\sum_{n \text{ odd}} (A_{n+1,k+1} + A_{n-1,k+1}) \frac{z^n}{n} = \frac{2^k}{k} \frac{z^k}{(1+z^2)^k}. \quad \text{(D.12)}$$

Using these expressions, we can again resum on $n$ to find

$$-\sum_{k\,\text{odd}} \int_p \frac{-2}{k} \left( \frac{2\Phi m}{p^2 + m^2 + \Phi^2} \right)^k = -\int_p \log \left( \frac{p^2 + (m+\Phi)^2}{p^2 + (m-\Phi)^2} \right). \tag{D.13}$$

Finally, after adding all the contributions in Eqs. (D.7), (D.10) and (D.13) together, we find considerable simplifications such that the final result gets the following simple form

$$\frac{V_{\text{eff}}(\Phi)}{N} = \frac{\Phi^2}{2g^2} - 2 \int_p \log \left( \frac{p^2 + (m+\Phi)^2}{p^2 + m^2} \right). \tag{D.14}$$

Once we have this generic result, we need to consider the case with several competing condensation channels $\Phi \mapsto \boldsymbol{\Phi} = (A, \Sigma, \Pi)$, and a specific twisted Wilson mass regularization. If we still consider a continuum QFT, but include the interaction term having full SO(5) symmetry, the expression for the effective potential generalises to

$$\frac{V_{\text{eff}}(A, \Sigma, \Pi)}{N} = \frac{1}{2g^2}(A^2 + \Sigma^2 + \Pi^2) + \sum_n \frac{1}{n} \int_p \text{tr} \left( \frac{-i\boldsymbol{\gamma} \cdot A - i\gamma_3 \Sigma_1 - i\gamma_5 \Sigma_2 - \Pi}{ip^\mu \gamma_\mu + im_1 \gamma_3 + im_2 \gamma_5} \right)^n. \tag{D.15}$$

Equations (D.2) and (D.3) are replaced by

$$\frac{1}{2g^2}(A^2 + \Sigma^2 + \Pi^2) + \sum_n \frac{1}{n} \int_p \left( \frac{1}{p^2 + m_1^2 + m_2^2} \right)^n \text{tr} I_n, \tag{D.16}$$

where we have introduced

$$I_n = -2(\boldsymbol{p} \cdot A + m_1 \Sigma_1 + m_2 \Sigma_2) I_{n-1} - (A^2 + \Sigma^2 + \Pi^2)(p^2 + m_1^2 + m_2^2) I_{n-2}. \tag{D.17}$$

It is now straightforward to repeat steps (D.7-D.13). After adding all the contributions, instead of Eq. (D.14), we find

$$\frac{V_{\text{eff}}}{N} = \frac{1}{2g^2}\boldsymbol{\Phi}^2 - 2 \int_p \log \left( \frac{p_0^2 + (\boldsymbol{p} + A)^2 + \boldsymbol{m}^2(\Sigma) + \Pi^2}{p_0^2 + \boldsymbol{p}^2 + \boldsymbol{m}^2(0)} \right), \tag{D.18}$$

where we have introduced $\boldsymbol{m}(\Sigma) = (m_1 + \Sigma_1, m_2 + \Sigma_2)$. At this point, in order to take into account the twisted-mass Wilson regularization, where we need to substitute $p = (p_0, \boldsymbol{p}) \mapsto (k_0, \hat{\boldsymbol{k}})$, $\boldsymbol{m}(\Sigma) \mapsto \boldsymbol{m}(\boldsymbol{k}, \Sigma)$, and substitute the momentum integrals by the corresponding mode sums. In this way, we arrive at Eq. (61) of the main text.

Finally we discuss incorporation of a further interaction between fermion charge densities mediated by an auxiliary $A_0$, whereupon Eqn. (D.15) is supplemented by terms

$$\frac{A_0^2}{2g'^2} + \sum_n \frac{1}{n} \int_p \text{tr} \left( \frac{-i\gamma_0 A_0}{ip^\mu \gamma_\mu + im_1 \gamma_3 + im_2 \gamma_5} \right)^n. \tag{D.19}$$

The choice $g'^2 = g^2$ yields full SO(6) symmetry. The algebra involving $\gamma_0$ is identical to that for $\boldsymbol{\gamma}$, and the one-loop contributions yield

$$-2 \int_p \log \frac{[(p_0^2 + A_0)^2 + P^2][(p_0^2 - A_0)^2 + P^2]}{(p^2 + Q^2)^2}, \tag{D.20}$$

where $Q^2 = \boldsymbol{p}^2 + \boldsymbol{m}^2(0)$, and $P^2 = (\boldsymbol{p} + A)^2 + \boldsymbol{m}^2(\Sigma) + \Pi^2$. Careful integration over $p_0$ with a UV cutoff $\Lambda$ show that all dependence on $A_0$ is $O(\Lambda^{-1})$ and hence vanishes as $\Lambda \to \infty$. As a consequence, the minimum of $V_{\text{eff}}$ always lies at $A_0 = 0$ and it is therefore safe to ignore condensation in this channel at half-filling.

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
