# Peer review of "A higher-order topological twist on cold-atom SO($5$) Dirac fields"

_SciPost Physics, doi:SciPost Phys. 17, 003 (2024)_

## Round 1 · Referee Report · Anonymous (Referee 1) · 2023-11-1

Report

In the manuscript "A higher-order topological twist on cold-atom SO(5) Dirac fields", the authors present a proposal to implement a fermionic quantum field theory with SO(5) symmetry in ultracold atom gases trapped in optical lattices. The theory behind this model is discussed with clarity in detail, and the authors focus on two important aspects: the relationship of their model with high-order topological insulators and the effects of the SO(5) symmetric interactions that extend the non-interacting picture.

The results presented are interesting and relevant for both the study of HOTIs and quantum simulations of field theories. However, there are weaknesses that the authors must address.

The first concerns the "hidden SO(5) symmetry". Usually, speaking about an SO(5) symmetry, one refers to a model which is symmetric under the full group, as, for instance, the field theory in Eq. 4. At the beginning of section 5, instead, the authors write that: "We argued previously that the HOTI groundstate is protected by a hidden SO(5) symmetry and, thus, should be robust under symmetric perturbations". I think that the symmetries protecting the HOTI are discrete symmetries, and that the robustness of the HOTI phase is much stronger. In particular, I tend to think that perturbations which are SO(5) symmetric are very artificial and not realistic in the ultracold atom system on the lattice. Indeed the HOTI phase of the non-interacting Hamiltonian 29 does not require a full SO(5) symmetry. Therefore I find the previous statement misleading. In general, I do not think the SO(5) symmetry plays an important role in defining the HOTI phase, as also showed in Fig.4, but only a small subset of discrete symmetries. A similar statement appears in the abstract, and, again, in my opinion it is confusing. And I find it confusing to refer to "a hidden SO(5) symmetry" (as in the abstract) because, if I understand correctly, this symmetry is just a discrete Pi/2 rotation in real space complemented with the specific transformation in Eq. 35, without any of the complexity of the continuous group. Therefore I would refrain from speaking about a "hidden SO(5) symmetry" to refer just to a specific rotation.

A second important point, in my opinion, is related to the experimental proposal. The paper lacks any concrete proposal about how to detect the HOTI phase in experiments or how to observe the main features of the model. I think this is a fundamental point to address in detail. It is crucial, for an experimental proposal as the one presented, to discuss how to validate its results.

Additionally, I also think that another important and related element that has not been discussed is how to obtain well-defined surfaces and corners in the optical lattice, and to discuss, for instance, what can happen to the corner topological modes in cases in which sharp surfaces are replaced, instead, by position-dependent confining potentials.

There are also minor points I invite the authors to consider:

1) If the authors wish, they could mention that also Weyl semimetals offer a potential example of 3D systems whose low-energy physics is well captured by relativistic (but not interacting) Dirac fields. 2) $t$ is used both for time, and the tunneling amplitudes in Eqs. 12 and 13. I suggest changing the font or the notation. 3) In the definition of $U_0$ and $U_{F_t}$, what is $k$? I am a bit puzzled because the densities are taken in real space on the lattice, and I find it misleading having an onsite density-density interaction that depends on momentum. 4) The authors write: "At the level of the twisted-mass free Hamiltonian (11), one sees that $\beta H_0(k) \beta = −H_0(k)$. This corresponds to the AIII class in the classification of topological insulators". I tend to disagree with this statement. When we consider the Hamiltonian 29, we see that, besides this sublattice chiral symmetry, the model displays also a particle-hole symmetry given by $\sigma_z \otimes \sigma_x$ and a time-reversal symmetry given by $\sigma_x \otimes \sigma_x$ (following the standard classification of topological insulators ans superconductors). Therefore I would say that the topological class is BDI instead of AIII. Furthermore, for the sake of clarity, I would consider these as non-spacial symmetries rather than global symmetries, to avoid confusion with the unitary global symmetries discussed in the work.

Finally, there are several typos to be corrected:

page 5: "lea to" -> "lead to" page 5: tau -> \tau caption of Fig. 1: "decouple"->"decoupled" page 9: "micorsocopic"->"microscopic" page 11: "of second order"->"of second order" page 14: "pseudo-sclarar"->"pseudo-scalar" page 17: "the higher-order topological invariant an the corner modes cannot coexist" page 18: "spin-3/3 Fermi gases" page 18: "it woudl"

In conclusion, I think that a revision is needed before considering this work for publication. In particular I invite the authors to carefully explain what they mean by "hidden SO(5) symmetry", or, even better, to avoid using this nomenclature which I find very confusing for a discrete symmetry. I also think that the discussion about the experimental proposal must be strengthened by discussing the possible signatures of the discussed phases. Without this aspect, I find the discussion on the experimental implementation of this model quite incomplete.

  • validity: -
  • significance: -
  • originality: -
  • clarity: -
  • formatting: -
  • grammar: -

Author:  Alejandro Bermudez  on 2024-03-27  [id 4378]

(in reply to Report 1 on 2023-11-01)

%%%%%%%%%%%%%%%%%%%%%%%%%%% % Answer to the 1st Referee %%%%%%%%%%%%%%%%%%%%%%%%%%%

We would like to thank the Referee for the very detailed report, and for making some specific recommendations that have helped us to clarify certain points. We were very pleased to read in the report that we managed to present our results with clarity and detail, and that these were indeed interesting an relevant not only for the study of HOTIs, but more generally for the research on quantum simulations of field theories. We now address in detail the points raised in the report:

I) Comment: The first concerns the "hidden SO(5) symmetry". Usually, speaking about an SO(5) symmetry, one refers to a model which is symmetric under the full group, as, for instance, the field theory in Eq. 4. At the beginning of section 5, instead, the authors write that: "We argued previously that the HOTI groundstate is protected by a hidden SO(5) symmetry and, thus, should be robust under symmetric perturbations". I think that the symmetries protecting the HOTI are discrete symmetries, and that the robustness of the HOTI phase is much stronger. In particular, I tend to think that perturbations which are SO(5) symmetric are very artificial and not realistic in the ultracold atom system on the lattice. Indeed the HOTI phase of the non-interacting Hamiltonian 29 does not require a full SO(5) symmetry. Therefore I find the previous statement misleading. 
In general, I do not think the SO(5) symmetry plays an important role in defining the HOTI phase, as also showed in Fig.4, but only a small subset of discrete symmetries. A similar statement appears in the abstract, and, again, in my opinion it is confusing. And I find it confusing to refer to "a hidden SO(5) symmetry" (as in the abstract) because, if I understand correctly, this symmetry is just a discrete Pi/2 rotation in real space complemented with the specific transformation in Eq. 35, without any of the complexity of the continuous group. Therefore I would refrain from speaking about a "hidden SO(5) symmetry" to refer just to a specific rotation.

Answer: We thank the Referee for this detailed comment, and for explaining the reasons that can potentially raise a misunderstanding when using the words ‘hidden SO(5) symmetry’ in the text. We fully agree that the relevant symmetry for the HOTi and all of the underlying physics discussed in this work is a discrete SO(5) rotation, not the full continuous symmetry group. The reason why we used ‘hidden SO(5) symmetry’ comes from the fact that, in the cold-atom community, there are previous works that focus on the appearance of this symmetry in spin-3/2 ultra cold atomic gases, as we have discussed and referenced in the text. However, it is true that, when including the additional terms that are responsible for the HOTI physics, the continuous symmetry is explicitly broken, and it is only a discrete SO(5) rotation that survives and is responsible for the symmetry protection of the HOTi phase.

In order to avoid any possible misunderstandings, we have restrained from using ‘hidden SO(5) symmetry’ in the amended version of the manuscript. We have substituted ‘hidden SO(5) symmetry’ by ‘discrete SO(5) rotation’ everywhere in the hope that it will not throw the potential readers into confusion. We thank the Referee for raising this important point.

II) Comment: A second important point, in my opinion, is related to the experimental proposal. The paper lacks any concrete proposal about how to detect the HOTI phase in experiments or how to observe the main features of the model. I think this is a fundamental point to address in detail. It is crucial, for an experimental proposal as the one presented, to discuss how to validate its results. Additionally, I also think that another important and related element that has not been discussed is how to obtain well-defined surfaces and corners in the optical lattice, and to discuss, for instance, what can happen to the corner topological modes in cases in which sharp surfaces are replaced, instead, by position-dependent confining potentials.

Answer: We thank the Referee for raising this point, which we have addressed in the amended version of the manuscript. We agree that a detailed discussion about the detection, including further numerical simulations for the specific model hereby studied, would be an additional value of the manuscript. However, we also stress that it would go beyond the original scope of this work, and make the manuscript even longer. Therefore, we have searched for a middle ground, addressing the definition of boundaries/corners and the detection and the in the new Sec. IV C. Here, we provide key references to quantum gas microscopes, and more recent results on the use of programmable optical potentials to create sharp boundaries in the optical lattices. We also comment on possible manifestations of the corner modes via adiabatic Thouless pumping, as well as the experimental inference of bulk entanglement by the reconstruction of the entanglement spectrum, as shown in recent works that are now cited in our amended text.

We now address the additional minor points also raised by the Referee: 1) If the authors wish, they could mention that also Weyl semimetals offer a potential example of 3D systems whose low-energy physics is well captured by relativistic (but not interacting) Dirac fields. Answer: We have introduced a sentence in the introduction, together with a couple of references. 2) t is used both for time, and the tunneling amplitudes in Eqs. 12 and 13. I suggest changing the font or the notation. is used both for time, and the tunneling amplitudes in Eqs. 12 and 13. I suggest changing the font or the notation. Answer: We have introduced.a sentence below Eq. (13) to avoid this possible confusion, emphasizing that only Euclidean time \tau will appear later on. 
3) In the definition of U0 and UFt, what is k ? I am a bit puzzled because the densities are taken in real space on the lattice, and I find it misleading having an onsite density-density interaction that depends on momentum. Answer: The $k$ that appears in both expressions is the laser wave vector. We have added a subscript $k_L$ to avoid any possible confusion. 
4) The authors write: "At the level of the twisted-mass free Hamiltonian (11), one sees that βH0(k)β=−H0(k). This corresponds to the AIII class in the classification of topological insulators". I tend to disagree with this statement. When we consider the Hamiltonian 29, we see that, besides this sublattice chiral symmetry, the model displays also a particle-hole symmetry given by σz⊗σx and a time-reversal symmetry given by σx⊗σx (following the standard classification of topological insulators ans superconductors). Therefore I would say that the topological class is BDI instead of AIII. Furthermore, for the sake of clarity, I would consider these as non-spacial symmetries rather than global symmetries, to avoid confusion with the unitary global symmetries discussed in the work. Answer: We thank the Referee for bringing this comment. We have changed our notation, and now refer to ’spatial and non-spatial symmetries’, as suggested. Regrading the first point, we thank the referee for noticing this mistake. We had been previously working with a different discretization in which $sin(k_ia) \leftrightarrow cos(k_ia)$, such that the explicit particle-hole and time-reversal symmetry where broken unless the bare masses $\tilde{m}_1=\tilde{m}_2=0$. However, for the current choice of the manuscript, the Referee is correct that this should correspond to a BDI class with C=σz⊗σxComplex_conjugation and T=σx⊗σxComplex_conjugation. We have corrected this on the amended manuscript. The referee has also spotted the following typos page 5: "lea to" -> "lead to"
page 5: tau -> \tau
caption of Fig. 1: "decouple"->"decoupled"
page 9: "micorsocopic"->"microscopic"
page 11: "of second order"->"of second order"
page 14: "pseudo-sclarar"->"pseudo-scalar"
page 17: "the higher-order topological invariant an the corner modes cannot coexist"
page 18: "spin-3/3 Fermi gases"
page 18: "it woudl" Which we have corrected. We would like to thank the Referee for all the helpful suggestions, which have helped us to improve the presentation of our results, and correct possible sources of confusion. Sincerely yours, The authors

---

## Round 1 · Referee Report · Anonymous (Referee 2) · 2023-11-27

Strengths

1- Interesting model discussed theoretically in detail. 2- Model has contributions to multiple disciplines, such as with HOTI phases and with non-trivial QFTs.

Weaknesses

1- No details on how to experimentally observe key features of the model. 2- Many grammatical errors.

Report

In the paper titled 'A Higher-Order Topological Twist on Cold-Atom SO(5) Dirac Fields,' the authors explore the behaviour of atoms confined to a two-dimensional Raman lattice, serving as a model for 4-Fermi interactions with a 'hidden' SO(5) symmetry encoded in the topology. The authors present a compelling theoretical framework with broad implications across disciplines and provide insights into experimentally realizing the proposed two-dimensional lattice. However, the manuscript falls short in suggesting methods for detecting key features of the model. Including such suggestions would significantly enhance the manuscript by bridging the theoretical framework with practical applications.

Moreover, the manuscript contains numerous spelling mistakes that warrant attention. I strongly encourage the authors to thoroughly review their work, paying particular attention to subtle errors that may be present in equations.

Before recommending this paper for publication, I would like the see these points addressed to hopefully further strengthen the manuscript's impact.

Requested changes

The following grammatical mistakes must be addressed:

1- Page 2, 'Moreover, this large N techniques can be readily used...' should say these, not this.

2- In the description of Figure 1, '... the inset (b) describes...' should say (c) instead. And mistake with making letters bold '...with those of bf (b) lead to flat bands.'

3- Page 5, '... can lea to a non-zero topological invariant', lea should be 'lead'.

4- Page 5, '...yielding a a 3-dimensional...' double 'a'.

5- Page 6, 'where the later coincide with the expressions in Eq. (14).' should say 'latter'?

6- Page 6, '...in light of the definition of the adjoint operator below Eq. (19).', should say Eq. (10).

7- Page 7, '...and in the internal electronic internal state given...', internal put twice.

8- Page 7, '...since the spinor components are only two.', should be 'since there are only two spinor components'?

9- Page 7, 'The only caveat is that we should consider other atomic species in which...', should 'other' say 'only'?

10- Page 9, '...each Raman beam to independently assists one single...', should be 'assist'.

11- Page 9, '...relevant dimensionless parameter that appear in the phase diagram...', 'parameters'.

12- Page 11, '...this phase should be a HOTI od second order...', should say 'of'.

13- Page 12, 'Chern insulator (C6) an the mass matrix (C4),', should say 'and'.

14- Page 12, '...Wilson loop associated to such winding number.', 'such a'.

15- Page 14, '...can be interpret as a specific SO(6)...' 'interpreted'.

16- Page 17, '...invariant an the corner modes...' 'in'.

17- Page 18, '...in Figs. 5 and6 for various...' '5 and 6'.

18- Page 18, '...it woudl be very ineteresting...' 'would'.

19- Page 18, '...for a different alkaline-earth 19 atoms such as...' 'for different'.

20- Page 20, '...leads to an effective low-energy action action that...' says action twice.

21- Page 21, '...or an combination of an odd number of them...', first 'an' should be 'a'.

22- Page 22, 'will be latter identified with' 'later'.

23- Page 23, 'we can again resum on n to' 'resume'.

24- Page23, Sigma subscripts of n odd/k odd in (D13/D14) don't look right.

25- Additionally, providing further comments on how to practically realize the proposed results, as I believe this will benefit the article and give it a wider impact.

  • validity: high
  • significance: good
  • originality: high
  • clarity: high
  • formatting: excellent
  • grammar: below threshold

Author:  Alejandro Bermudez  on 2024-03-27  [id 4379]

(in reply to Report 2 on 2023-11-27)

%%%%%%%%%%%%%%%%%%%%%%%%%%%
% Answer to the 2nd Referee
%%%%%%%%%%%%%%%%%%%%%%%%%%%

We would like to thank the Referee for the assessment of our work, and for considering that it presents a compelling framework with broad multidisciplinary implications. Regarding the suggestions:

I) Comment: However, the manuscript falls short in suggesting methods for detecting key features of the model. Including such suggestions would significantly enhance the manuscript by bridging the theoretical framework with practical applications.

Answer: We thank the Referee for raising this point, which we have addressed in the amended version of the manuscript. We agree that a detailed discussion about the detection, including further numerical simulations for the specific model hereby studied, would be an additional value of the manuscript. However, we also stress that it would go beyond the original scope of this work, and make the manuscript even longer. Therefore, we have searched for a middle ground, addressing the definition of boundaries/corners and the detection and the in the new Sec. IV C. Here, we provide key references to quantum gas microscopes, and more recent results on the use of programmable optical potentials to create sharp boundaries in the optical lattices. We also comment on possible manifestations of the corner modes via adiabatic Thouless pumping, as well as the experimental inference of bulk entanglement by the reconstruction of the entanglement spectrum, as shown in recent works that are now cited in our amended text.

II) Comment: Moreover, the manuscript contains numerous spelling mistakes that warrant attention. I strongly encourage the authors to thoroughly review their work, paying particular attention to subtle errors that may be present in equations.

Answer: We agree with the Referee, as we have found additional spelling mistakes in the previous version of the manuscript. We have now checked this in detail, and are confident that the current version of the manuscript has now been improved.

We would like to thank the Referee for the assessment of our work.
Sincerely yours,
The authors

---

## Round 2 · Referee Report · Anonymous (Referee 2) · 2024-4-5

Report

I am happy with the changes made and would like to recommend this paper for publication.
  • validity: -
  • significance: -
  • originality: -
  • clarity: -
  • formatting: -
  • grammar: -

Author:  Alejandro Bermudez  on 2024-05-10  [id 4481]

(in reply to Report 1 on 2024-04-05)
Category:
answer to question

%%%%%%%%%%%%%%%%%%%%%%%%%%%%%%%%%%%%%%%%%%%%%%%%
% Resubmission of “A higher-order topological twist on cold-atom SO(5) Dirac fields”
%%%%%%%%%%%%%%%%%%%%%%%%%%%%%%%%%%%%%%%%%%%%%%%%

We would like to thank the Referee for assessing our first resubmission, and for considering that the changes performed sufficed to merit publication in SciPost.

Sincerely yours,

The authors

---

## Round 2 · Referee Report · Anonymous (Referee 1) · 2024-4-8

Strengths

The results presented are interesting and relevant for both the study of HOTIs and quantum simulations of field theories.

Weaknesses

The discussion on the experimental protocols to detect the non-trivial phases of the addressed systems is weak and generic.

Report

The authors improved the clarity of their exposition.
However, I ask a further effort to extend Sec. IVC which, in my opinion, is not yet sufficient to convey a realistic idea about the observation of the boundary modes. Using the authors' words, I may say that I am not satisfied by the "middle ground" they have chosen.

In particular, the authors mention that "localized boundary modes, such as corner modes, can be directly detected in real space by locally resolving the atomic density using a quantum gas microscope". However, a straightforward use of a quantum gas microscope (applied to the ground state of the fermionic system) would not distinguish in general the contribution of bulk, edge or corner modes, and would return the overall density resulting from all these contributions.

How can the local density discriminate between the trivial and HOTI phases? Is there a difference in the corner density one expects in the two phases for sharp boundaries? Can the authors specify more explicitly whether this happens in a specific limit, or, in general, there is a discontinuity at the phase transition? Is there a way of doing spectroscopy to obtain a local density of states?

Or are the authors thinking about some specific out-of-equilibrium protocol as in Ref. 207?

Also the comment on the entanglement spectrum is too vague. Ref. 210 concerns ions used as qubits, and not atoms. And I tend to think that the density matrices that one has to calculate to obtain indications about the HOTI phases in this ultracold atom setup have almost nothing in common with that work (different physical platform and different basis: in the HOTI I think one has to consider the occupation numbers of the particles, in Ref. 210 it was the inner degrees of freedom of the ions).

In conclusion, I think that the manuscript will be suitable for publication in SCIPOST Physics only after the authors extend Sec. IVC by outlining physical protocols for the detection of the HOTI phases in a more detailed and rigorous way.

Additional minor remarks:

In this webpage, the first line of the abstract is incomplete. The manuscript is correct.

In page two " ‘discrete SO(5) rotation " -> " ‘discrete SO(5) rotation' "

Concerning the notation "t" used for both time and the tunneling, I still find that not all the issues have been solved. "t" is introduced as a tunneling amplitude in Eq. 12. However, in the exponents of Eqs. 25 and 26, "t" is still labelling the time. I ask the authors a further effort to avoid this ambiguity in the notation.

Requested changes

Extension of Sec. IVC in a more rigorous and detailed form.

  • validity: -
  • significance: -
  • originality: -
  • clarity: -
  • formatting: -
  • grammar: -

Author:  Alejandro Bermudez  on 2024-05-10  [id 4482]

(in reply to Report 2 on 2024-04-08)
Category:
answer to question

%%%%%%%%%%%%%%%%%%%%%%%%%%%%%%%%%%%%%%%%%%%%%%%%
% Resubmission of “A higher-order topological twist on cold-atom SO(5) Dirac fields”
%%%%%%%%%%%%%%%%%%%%%%%%%%%%%%%%%%%%%%%%%%%%%%%%

We would like to thank the Referee for going through the changes we implemented in our first resubmissions nd for reading critically our new subsection about the possible detection. We are happy to see that the Referee found the majority of the changes performed satisfactory, and that these “improved the clarity of their exposition”.

Regarding the Referee’s concern about detection of topological edge states in cold-atom experiments, we have now extended the “experimental detection” section to make the explanation more detailed. The referee is concerned in particular that the use of a quantum gas microscope “would not distinguish in general the contribution of bulk, edge or corner modes, and would return the overall density resulting from all these contributions.” We think this would not be the case, since the presence of edge/corner states creates an excess of atomic density at the boundaries of the topological phase as compared to the trivial phase. This was indeed measured experimentally in a recent cold-atom experiment where the 1D Haldane phase was realized using a Fermi-Hubbard system [Nature 606, 484–488 (2022)], and the same is true for corner states in 2D [Phys. Rev. B 102, 041126(R) (2020)]. We now mention this explicitly in the manuscript.

Regarding the extraction of the entanglement spectrum, we agree with the referee that the reference we mentioned in our first resubmission addresses trapped-ion systems. We have now updated the reference with a protocol designed to extract the entanglement spectrum in cold-atom systems, which can be done by measuring local densities and currents.

We believe that this section now contains a reasonable discussion of possible detection methods. Even if we agree with the Referee that this could be extended further, even performing numerical simulations to test the ideas for our specific model, this would require a considerable amount of work and go beyond the scope of the manuscript, in our most sincere opinion. We believe that our work already contains several novel results and interesting connections between four-Fermi relativistic lattice field theories and interacting higher-order topological phases, which are independent of possible methods for experimental detection.

The Referee has also noticed a typo and an inconsistency in the use of $t$ for both real time and tunneling strength. We have corrected these two issues in the amended version of the manuscript (see the highlighted red changes)

We would like to thank the Referee for the overall assessment of our work.
Sincerely yours,
The authors

Anonymous on 2024-05-13  [id 4483]

(in reply to Alejandro Bermudez on 2024-05-10 [id 4482])
Category:
reply to objection

The updated version (v3) of Sec. IVc is more convincing and complete than the previous. I am now convinced that this work provides a genuine contribution to the field, relevant also for future experiments. For these reasons I support its publication in SciPost Physics.

---

## Round 2 · List of Changes

%%%%%%%%%%%%%%%%%%%%%%%%%%% % List of changes %%%%%%%%%%%%%%%%%%%%%%%%%%%

  • We have corrected several typos spotted by the 1st Referee (red in the marked pdf).

  • Following the suggestion of the 2nd Referee, we have checked the grammar and corrected additional typos (red in the marked pdf)

  • Following the suggestion of the 1st Referee, we have avoided using the adjective ‘hidden’ when referring to the SO(5) protecting symmetry, and used instead ‘discrete SO(5) rotation’, emphasizing that this describes the invariance of the model under a discrete subgroup of SO(5).

  • We have corrected the mistake regarding the connection to the AIII class versus the BDI class at several places of the manuscript (see red changes in the market manuscript)

  • We have included a new subsection IV.C to discuss the detection in ultra cold atomic gases, and the possibility of inducing sharp boundaries with optical potentials.

---

## Round 3 · Author Response

%%%%%%%%%%%%%%%%%%%%%%%%%%%%%%%%%%%%%%%%%%%%%%%%
% Resubmission of “A higher-order topological twist on cold-atom SO($5$) Dirac fields”
%%%%%%%%%%%%%%%%%%%%%%%%%%%%%%%%%%%%%%%%%%%%%%%%

Dear Editors of SciPost,

We would like to thank you for giving us the opportunity to address the comments raised by the second Referee in the second round of referral. We have addressed them, performing the suggested changes. Given also that the the first Referee already recommended our first revised work to be published, we believe that our amended manuscript is now ready to be published in SciPost.

Please find below a list of changes, together with our detailed answer to the Referees.

---

## Round 3 · List of Changes

%%%%%%%%%%%%%%%%%%%%%%%%%%% % List of changes %%%%%%%%%%%%%%%%%%%%%%%%%%%

  • We have corrected several typos spotted by the 1st Referee (red in the marked pdf).

  • Following the suggestion of the Referee, we have corrected typos and avoided inconsistencies in the notation regarding tunneling strengths an real time (red in the marked pdf)

  • Following the suggestion of the Referee, we have extended the ‘experimental detection subsection IV.C ’.

---

## Editorial Decision

published